# An RNA-binding protein acts as a major post-transcriptional modulator in *Bacillus anthracis*

Hualiang Pi[1,2], Andy Weiss[1,2], Clare L. Laut[1,2], Caroline M. Grunenwald [1,2], Hannah K. Lin[3], Xinjie I. Yi[3], Devin L. Stauff[3] & Eric P. Skaar [1,2✉]

HitRS is a two-component system that responds to cell envelope damage in the human pathogen *Bacillus anthracis*. Here we identify an RNA-binding protein, KrrA, that regulates HitRS function by modulating the stability of the *hitRS* mRNA. In addition to *hitRS*, KrrA binds to over 70 RNAs and, directly or indirectly, affects the expression of over 150 genes involved in multiple processes, including genetic competence, sporulation, RNA turnover, DNA repair, transport, and cellular metabolism. KrrA does not exhibit detectable nuclease activity in vitro, and thus the mechanism by which it modulates mRNA stability remains unclear.

[1] Department of Pathology, Microbiology, & Immunology, Vanderbilt University Medical Center, Nashville, TN, USA. [2] Vanderbilt Institute for Infection, Immunology, and Inflammation, Vanderbilt University, Nashville, TN, USA. [3] Department of Biology, Grove City College, Grove City, PA, USA. ✉email: skaar@vumc.org

**B**acillus anthracis is a Gram-positive, spore-forming, facultative anaerobe, and the causative agent of the disease anthrax. The spore is the infectious form of this organism and can remain dormant and survive environmental extremes for decades[1,2]. B. anthracis has been considered one of the most serious bioterrorism threats[3], and it also causes endemic disease in certain regions of the world. It is estimated that 1.83 billion people and 1.1 billion livestock live within vulnerable regions of anthrax risk worldwide[4]. Anthrax presents as one of four forms of the disease: cutaneous, gastrointestinal, injectional, and inhalational. Inhalation anthrax is the most deadly form of the disease with mortality rates approaching 90% for untreated infections[5]. To survive and propagate within the hostile host environments, B. anthracis has evolved elaborate signal transduction systems involved in stress detection and detoxification known as two-component systems (TCSs)[6–11].

TCSs are a predominant strategy used by bacteria to coordinate detection of external cues to alterations in gene expression. A prototypical TCS consists of a membrane-bound histidine kinase (HK) and a cytoplasmic response regulator (RR)[12–14]. HKs sense a specific signal, autophosphorylate at a conserved histidine residue, and transfer the phosphate onto a conserved aspartate of the cognate RR. This phosphorylation event activates the RR, which often functions as a transcriptional regulator. The activated regulator binds to target promoters, alters gene expression, and modulates bacterial physiology to relieve cellular stress[12–14]. TCSs are widespread in bacteria but absent in humans and animals, making them potential targets for developing novel antibacterial agents[12–14].

In addition to being exquisitely regulated by phosphorylation, interactions between HK and RR pairs, and the second messenger cyclic-di-adenosine monophosphate[15], some TCSs are controlled by connector or modulator proteins that affect phosphorylation of either the HK or RR[16–18]. For instance, PmrD in Salmonella enterica is under control of the TCS PhoPQ in response to low magnesium levels. PhoPQ-activated PmrD in turn promotes phosphorylation of the RR of another TCS, PmrAB. This is illustrative of a physiological connection between two TCSs mediated by a connector protein[19,20]. In Bacillus subtilis, several proteins modulate signal tranduction of TCSs involved in sporulation such as Sda, KipI, and the Rap family of proteins, all of which affect TCS phosphorylation cascades[16]. These TCS modulators determine the timing and duration of TCS activation, which is critical to ensure the proper response to environmental stimuli. Such TCS modulators are often specific for an individual TCS, ensuring specificity of the response by preventing unwanted TCS cross-signaling. Furthermore, TCS modulators display very-low sequence identity, which makes it challenging to identify new TCS regulatory proteins based soley on primary structure[16].

B. anthracis encodes approximately 45 TCSs, including a heme-sensor system (HssRS) and a HssRS interfacing TCS (HitRS)[6–9,21]. These two TCSs cross-regulate and integrate TCS signaling in response to both cell envelope disruptions and heme intoxication[6]. HitS contains a 15-amino-acid sensor domain and has been characterized as an intramembrane-sensing HK[9,22,23]. This group of HKs detects signals within the membrane interface and is often involved in the response to cell envelope stress[23], which coincides with HitRS being activated by a variety of cell envelope-acting compounds[6,8]. We have recently uncovered the molecular determinants critical for activities intrinsic to HitRS signal transduction[9]. However, it remains unclear how HitRS signaling is regulated to ensure specificity of the cellular response in times of cell envelope damage.

In this study, a genetic selection strategy was employed to identify accessory proteins that are required for regulating the HitRS system. We identified a regulatory protein that we named KrrA (Kre-related RNA regulatory protein A), which plays an important role in HitRS activation through modulating mRNA stability of TCS gene transcripts. These data demonstrate that KrrA functions as an RNA binding protein (RBP) and plays an important post-transcriptional regulatory role in HitRS signaling. In addition, we integrated biochemical, genetic, and transcriptomic strategies to elucidate the underlying mechanism of KrrA-mediated RNA regulation. These results reveal a broader role for KrrA in the regulation of numerous mRNA targets involved in cellular processes such as genetic competence, germination, and sporulation, implicating KrrA as a critical global regulatory factor in B. anthracis and providing new insights into post-transcriptional gene regulation in bacteria.

## Results

**Identification of a TCS modulator critical for HitRS signaling.** To identify regulatory factors required for HitRS activation, a genetic selection strategy was employed. Briefly, B. anthracis WT strain Sterne harboring the erythromycin resistance gene ermC driven by the hitPRS promoter (WT P$_{hit}$ermC)[9], which is integrated into a pseudogene locus bas3009, was spread on media containing toxic levels of erythromycin and colonies that arose represented bacteria that acquired mutations constitutively activating the P$_{hit}$ promoter (Fig. 1A). As we have previously reported[9], the basal expression level of the hitPRS operon is very low in this strain, which makes this genetic selection a sensitive and powerful tool for detecting subtle changes related to HitRS activation. Twenty-one suppressors were isolated in this assay, all of which showed resistance to erythromycin and exhibited no mutations within hitRS. Whole-genome sequencing revealed that each isolated suppressor contained a frameshift mutation at one of five different locations within the same gene, bas3905 (Fig. 1B, Supplementary Fig. 1, and Supplementary Table 3). No other mutations were identified in these strains. The protein Bas3905 exhibits 60% sequence identity to the ComK repressor (Kre) from B. subtilis, which has been proposed to modulate the induction of the master regulator of genetic competence ComK[24], although the specific mechanism by which Kre acts is unknown. The B. subtilis kre mutant has enhanced transformation efficiency[24], and we observed that this phenotype is fully complemented by providing an inducible copy of B. anthracis bas3905 in the B. subtilis kre deletion mutant (Supplementary Fig. 2A), suggesting that Bas3905 performs a similar function. Thus we named this protein KrrA (Kre-related RNA regulatory protein A). To further confirm its role in genetic competence, a clean deletion mutant of krrA (ΔkrrA) was constructed and transformation efficiency was examined. B. anthracis ΔkrrA exhibited a significantly enhanced ability to uptake exogenous DNA compared to WT (Supplementary Fig. 2B). These data demonstrate that KrrA compensates for the loss of kre in B. subtilis and plays an important role in induced genetic competence in B. anthracis.

To test whether krrA deletion itself contributes to resistance to erythromycin, a growth assay was employed. ΔkrrA showed similar levels of erythromycin susceptibility compared to WT (Supplementary Fig. 2C), indicating that deletion of krrA alone has no effects on resistance to erythromycin. To further validate the genetic selection, a krrA null strain harboring P$_{hit}$ermC was constructed. As expected, this strain (ΔkrrA P$_{hit}$ermC) gained strong erythromycin resistance while the parental strain (WT P$_{hit}$ermC) was susceptible to erythromycin (Fig. 1C, D), suggesting that deletion of krrA leads to constitutive activation of the P$_{hit}$ promoter. This phenotype was substantially complemented in trans using a FLAG-tagged KrrA construct (ΔkrrA P$_{hit}$ermC pOS1.P$_{lgt}$krrA-3xFLAG) (Fig. 1C, D). Furthermore, erythromycin resistance was not observed in the triple-mutant

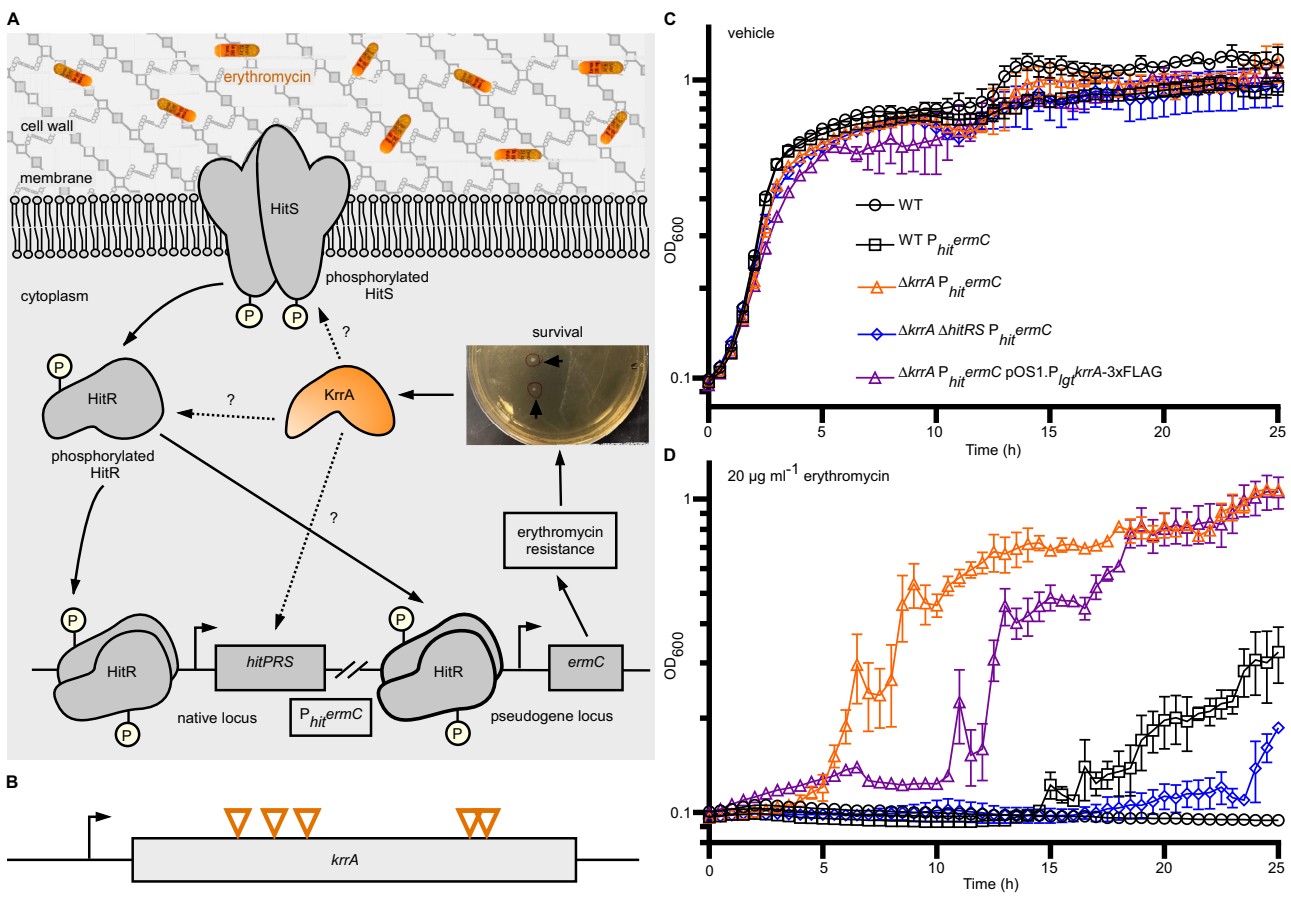

**Fig. 1 An unbiased genetic selection strategy identifies KrrA as a regulatory factor in HitRS signaling. A** Schematic of the genetic selection strategy: a strain containing *ermC* driven by a HitR promoter (P$_{hit}$*ermC*) was plated on medium containing toxic levels of erythromycin and colonies that arose represented bacteria that acquired mutations that constitutively activate the P$_{hit}$ promoter. **B** All 21 erythromycin-resistant suppressors isolated from genetic selections exhibit frameshift mutations at five different positions within *krrA*. **C, D** Growth kinetics of *B. anthracis* WT, WT P$_{hit}$*ermC*, Δ*krrA* P$_{hit}$*ermC*, Δ*krrA* Δ*hitRS* P$_{hit}$*ermC*, and the complemented strain Δ*krrA* P$_{hit}$*ermC* pOS1.P$_{lgt}$*krrA*-3xFLAG in vehicle (**C**) or 20 μg ml⁻¹ erythromycin (**D**) were monitored for 24 h. The experiments were conducted at least three times. Data shown are averages of three biological replicates (mean ± SD).

(Δ*krrA* Δ*hitRS* P$_{hit}$*ermC*) (Fig. 1C, D), suggesting that HitRS is required for the induction of erythromycin resistance. Altogether, these data indicate that KrrA functions as a TCS modulator and plays an essential regulatory role in HitRS signaling.

**The HitRS signaling system is finely tuned by KrrA through modulating mRNA stability.** We hypothesized that KrrA functions as a transcription factor that represses expression of the *hitPRS* operon. To test this, quantitative PCR (qPCR) was used to determine the mRNA levels of *hitP*, *hitR*, and *ermC* in four representative *krrA* suppressor mutants isolated from the genetic selection. VU0120205 ('205), a small synthetic compound identified as a HitRS inducer through a high-throughput screen[6], was used to evaluate the transcriptional changes under HitRS-activating conditions. However, no significant differences were observed under either vehicle (0.1% DMSO) or '205-treated conditions between any of the suppressor mutants and the parental strain (WT P$_{hit}$*ermC*) (Supplementary Fig. 3A–C). Similarly, deletion of *krrA* exhibited only minor effects (approximately twofold increase) on transcription of *hitP* or *hitR* under either vehicle (Supplementary Fig. 3D) or '205-treated conditions (Supplementary Fig. 3E). Since cross-regulation has been observed between HitRS and HssRS[6], the effects of *krrA* deletion on transcription of other TCS genes including *hssR*[6] and *edsR*[7] were evaluated under activating conditions for these two TCSs.

However, no significant differences were observed between WT and Δ*krrA* for all genes tested under these conditions (Supplementary Fig. 3D–G). Activation of HitRS and EdsRS was also examined through quantifying the promoter activity of P$_{hit}$ and P$_{eds}$ in WT and Δ*krrA* carrying a XylE reporter. Only approximately a twofold increase in P$_{hit}$ activity was observed in Δ*krrA* compared to WT under vehicle-treated conditions and this phenotype was diminished following '205 treatment (Supplementary Fig. 3H, I). In addition, *krrA* deletion showed no effects on EdsRS activation (Supplementary Fig. 3H, I). Together, these data suggest that KrrA does not play a major role in *hitPRS* transcription.

We then hypothesized that KrrA may alter the half-life of *hitPRS* transcripts. To test this, Northern blot analysis was carried out on both WT and Δ*krrA* in the presence or absence of '205. Consistent with our prior finding that the basal expression of *hitPRS* is very low[9], the *hitPRS* transcript was only detectable in Δ*krrA* treated with the inducer '205 while the housekeeping gene *16 S rRNA* showed similar abundance between the two strains under both conditions (Fig. 2A), indicating that KrrA plays a significant role in modulating *hitPRS* transcript stability.

Due to the low abundance of *hitPRS* transcript, an mRNA stability assay using a more sensitive detection method of qPCR was further employed. The transcript of *hitR* is more stable in the representative suppressor B26 compared to the parental strain (WT P$_{hit}$*ermC*). The half-life of *hitR* transcripts in B26 and the

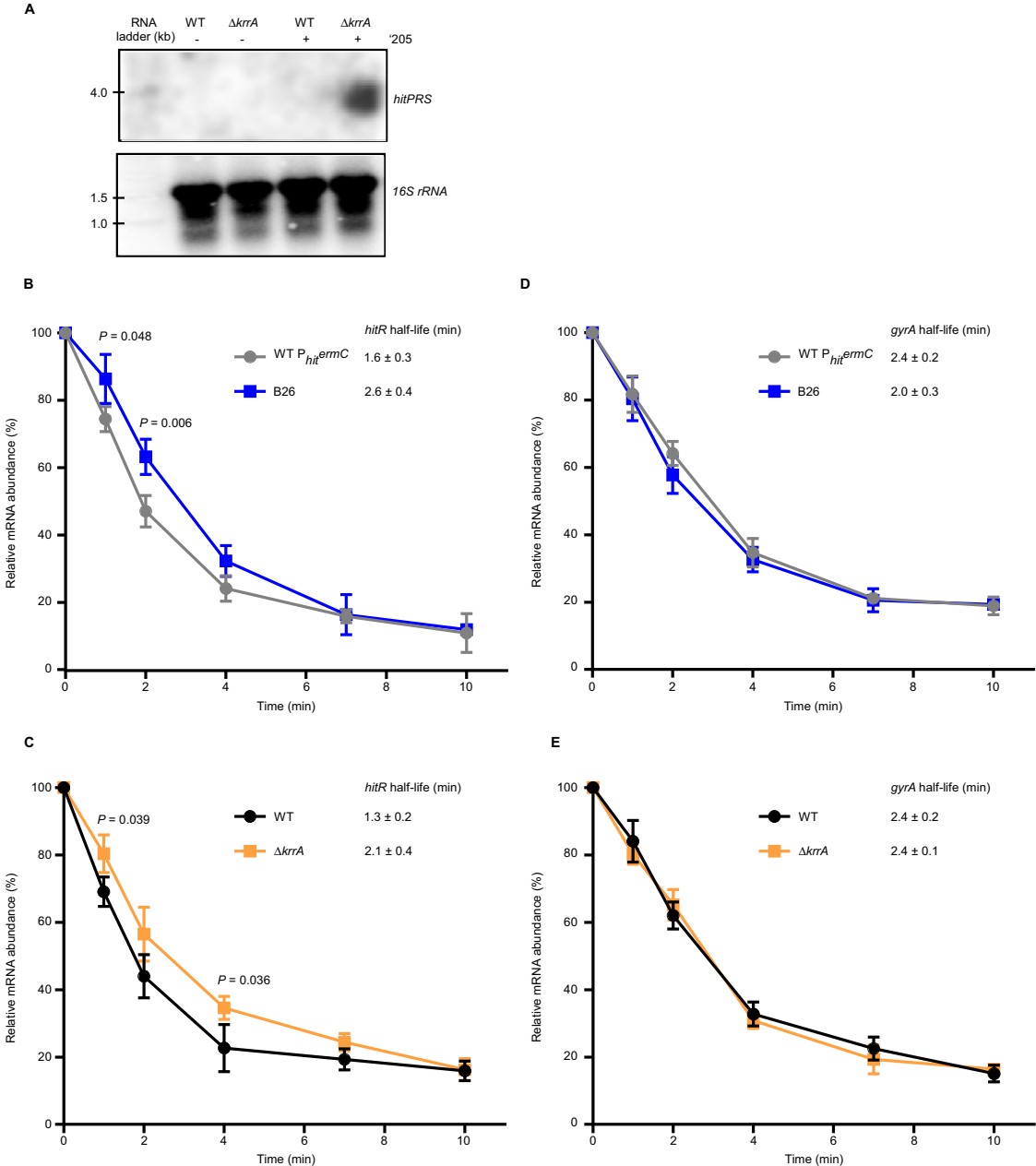

**Fig. 2 The HitRS signaling system is finely tuned by KrrA through modulating mRNA stability. A** To examine the effects of *krrA* deletion on the stability of *hitPRS* transcripts, northern blot analysis was carried out in WT and Δ*krrA* in the presence or absence of '205 using a pool of four sequence-specific probes against *hitPRS*. The housekeeping gene *16 S rRNA* serves as a sample loading control. The experiments were performed at least three times and representative images are shown. Source data are provided as a Source Data file. The mRNA stability of *hitR* (**B, C**) and *gyrA* (**D, E**) was examined in two sets of strains: *B. anthracis* WT P*hit*ermC and B26 suppressor mutant (**B, D**), and WT and Δ*krrA* (**C, E**). Relative abundance of *hitR* (**B, C**), or *gyrA* (**D, E**) was quantified using qPCR. The housekeeping gene *gyrA* serves as a negative control. The mRNA half-life was determined using a single exponential decay model. Data shown are three biological replicates (mean ± SD). Significant differences are determined by two-tailed *t* tests.

parental strain is 2.6 and 1.6 min, respectively (Fig. 2B). The mRNA stability of *hitR* between Δ*krrA* and WT was also compared, and as anticipated, *hitR* mRNA is more stable in Δ*krrA* compared to WT (Fig. 2C). Transcripts corresponding to the housekeeping gene *gyrA* served as a control in these experiments and did not show significant differences between these two sets of strains (Fig. 2D, E). HitS protein levels were then evaluated by western blotting and an increase in the intensity of HitS was detected in Δ*krrA* compared to WT, particularly under '205-treated conditions (Supplementary Fig. 4). Collectively, these

results demonstrate that HitRS signaling is finely tuned by KrrA through modulating the stability of *hitR* transcripts.

**KrrA has no detectable nuclease activity in vitro**. We next sought to define the molecular mechanism by which KrrA modulates *hitR* mRNA stability. This predicted cytosolic protein contains 152 amino acids with no known functional motifs or nucleic-acid-binding domains. Based on the impact of KrrA on the half-life of *hitR* transcripts, we hypothesized that KrrA may

function as a nuclease that plays a direct role in RNA degradation. To test this, recombinant KrrA was purified and both DNA and RNA nuclease activities of KrrA were examined in vitro. EcoRI, active in CutSmart buffer, served as a positive control for DNase activity, while RNase A, active in a phosphate buffer, was used as a positive control for RNase activity (Supplementary Fig. 5). No detectable DNA or RNA nuclease activity was observed with up to 1 or 10 μM of KrrA protein tested, respectively (Supplementary Fig. 5). These results suggest that KrrA does not degrade DNA or RNA directly in vitro and may require an unknown cofactor(s) for its activity.

**KrrA binds RNA directly in *B. anthracis* and a subset of the KrrA-RNA interactome is activated by HitRS stimulation.** Many RBPs contain no previously characterized RNA-binding domain[25], therefore we hypothesized that KrrA functions as an RBP with an as-yet-unidentified RNA-binding domain, and that KrrA binding to mRNA facilitates transcript processing and/or degradation. To begin to test this hypothesis, a FLAG-tagged KrrA driven by a constitutive promoter P$_{lgt}$ (pOS1.P$_{lgt}$krrA-FLAG) was constructed, and its functionality was validated using a growth assay (Fig. 1C, D). This construct was introduced into Δ*krrA*, and a CLIP assay (UV cross-linking RNA immunoprecipitation) was carried out to detect KrrA-interacting RNA complexes. The immunoprecipitated KrrA-FLAG fusion protein was detectable by immunoblotting using an anti-FLAG antibody (Fig. 3A), indicative of robust immunoprecipitation. The co-immunoprecipitated RNAs were subjected to radioactive labeling and detected by phosphorimaging even without cross-linking (Fig. 3A). As anticipated, cross-linking drastically enhanced the recovery of co-immunoprecipitated RNAs (Fig. 3A). These results demonstrate that KrrA directly binds RNA in vivo. We next hypothesized that KrrA binds *hitR* transcripts to facilitate *hitR* mRNA decay, resulting in constitutive activation of HitRS signaling in the absence of KrrA. To test this, the HitRS activator '205 was added to the growth medium to enrich *hitRS* transcripts. Indeed, the radioactive signal was considerably stronger upon HitRS stimulation (Fig. 3A), indicating that KrrA acts as an RBP and some of its interacting RNA targets are enriched following HitRS stimulation.

**KrrA-interacting RNA targets are involved in many cellular processes.** We hypothesized that KrrA specifically binds to a set of mRNA transcripts in vivo and impacts mRNA turnover. To test this, formaldehyde cross-linking RNA immunoprecipitation coupled with Illumina sequencing (fRIP-seq) was carried out to pinpoint the RNA targets of KrrA. Since KrrA-interacting RNA transcripts were enriched following HitRS stimulation (Fig. 3A), we expected to detect higher abundance of KrrA-targeted RNAs and/or additional targets following HitRS stimulation. Thus, two different growth conditions were implemented: LB and LB supplemented with '205. Two controls were included to ensure specificity: (i) a WT strain that does not harbor the FLAG-tagged KrrA construct as a nonspecific background control, and (ii) RNA extracted from WT cells harvested before immunoprecipitation, serving as an input RNA control. Overall, a total of 359 RNA transcripts were specifically enriched upon KrrA immunoprecipitation (fold enrichment ≥3; $P < 0.05$) under both vehicle and '205-treated conditions, excluding 38 transcripts identified from the untagged control samples (Fig. 3B). Approximately 40% of the candidate KrrA targets (138 out of 359) were present in both vehicle and '205-treated conditions, including *hitR* transcripts. As expected, more RNA targets were identified (181 out of 359) following '205 treatment, including *hitS* and *hitP* (Fig. 3B). A complete list of KrrA targets is included in Supplementary Data 1.

Given that Δ*krrA* exhibits an enhanced ability to uptake exogenous DNA following the induction of competence (Supplementary Fig. 2B), we postulated that KrrA might interact with transcripts that encode proteins important for genetic competence. ComFA is a ssDNA-dependent ATPase that forms a stable complex with ComFC. These two proteins are essential for DNA uptake during transformation[26]. Indeed, both transcripts in the *comF* operon were significantly enriched following KrrA immunoprecipitation under both vehicle and HitRS-activating conditions: 17–20-fold enrichment for *comFA* and 6–11-fold for *comFC* (Supplementary Data 1). These data suggest that KrrA modulates inducible genetic competence through interacting with transcripts involved in DNA transformation. RNA targets of KrrA are also involved in many other biological processes, including cellular metabolism, DNA repair, electron transfer, metal homeostasis, RNA turnover, sporulation, transport, and transcriptional regulation (Fig. 3C and Supplementary Data 1), indicating KrrA plays a broad regulatory role in bacterial physiology. Additionally, KrrA interacts with transcripts encoding 35 transcriptional regulators: 3 response regulators and 32 one-component regulators (Fig. 3C and Supplementary Data 1). This finding indicates that KrrA acts as a pleiotropic post-transcriptional RNA modulator that drives the transcriptional landscape of *B. anthracis* both directly through interacting with transcripts involved in key cellular processes and indirectly through modulating stability of mRNA encoding for transcription factors.

To further define the transcripts enriched following HitRS stimulation, the enrichment levels of 319 transcripts identified following '205 treatment were compared to those under vehicle-treated conditions. Three transcripts were most significantly enriched upon HitRS stimulation: *hitP*, *hitS*, and *bas5289*, which encodes an uncharacterized efflux transporter, with fold of enrichment being 243, 25, and 11, respectively (Fig. 3D and Supplementary Data 1). These data suggest that the strong signal observed in the CLIP assay (Fig. 3A) was primarily due to '205 stimulating *hitPRS* expression. In particular, the sequencing reads were substantially enriched in the entire *hitPRS* operon following '205 treatment and only *hitR* transcripts were enriched under vehicle-treated conditions (Fig. 3E, F), which was further validated by fRIP-qPCR (Fig. 3G). To examine whether KrrA directly binds to *hitR* transcript in vitro, an electrophoretic mobility shift assay (EMSA) was carried out using in vitro-transcribed and radioactively labeled RNA. KrrA-*hitR*-transcript complexes were observed with as low as 100 nM of KrrA, however, no complexes were detectable for the control RNA with up to 1 μM KrrA tested (Fig. 4), indicating that KrrA specifically binds to *hitR* transcript. Taken together, these data support a model whereby KrrA facilitates mRNA turnover through interacting directly with its targets such as *hitPRS*, suggesting a post-transcriptional regulatory role for KrrA in TCS signaling and likely other cellular processes.

**KrrA-mediated RNA regulation broadly affects the abundance of numerous transcripts.** To uncover the effects of KrrA regulation on gene transcription and define the bacterial response imposed by this regulation, a transcriptomic study was carried out in which *B. anthracis* Δ*krrA* was compared to WT in two growth conditions: LB in the presence or absence of '205. A total of 774 genes were affected by KrrA-mediated RNA regulation (Fig. 5A–C and Supplementary Data 2). Interestingly, the vast majority of affected genes showed increased expression levels in Δ*krrA* compared to WT (Fig. 5A, B), suggesting that KrrA facilitates RNA turnover for at least some of these transcripts. More than 50% of these transcripts (402 out of 774) were affected

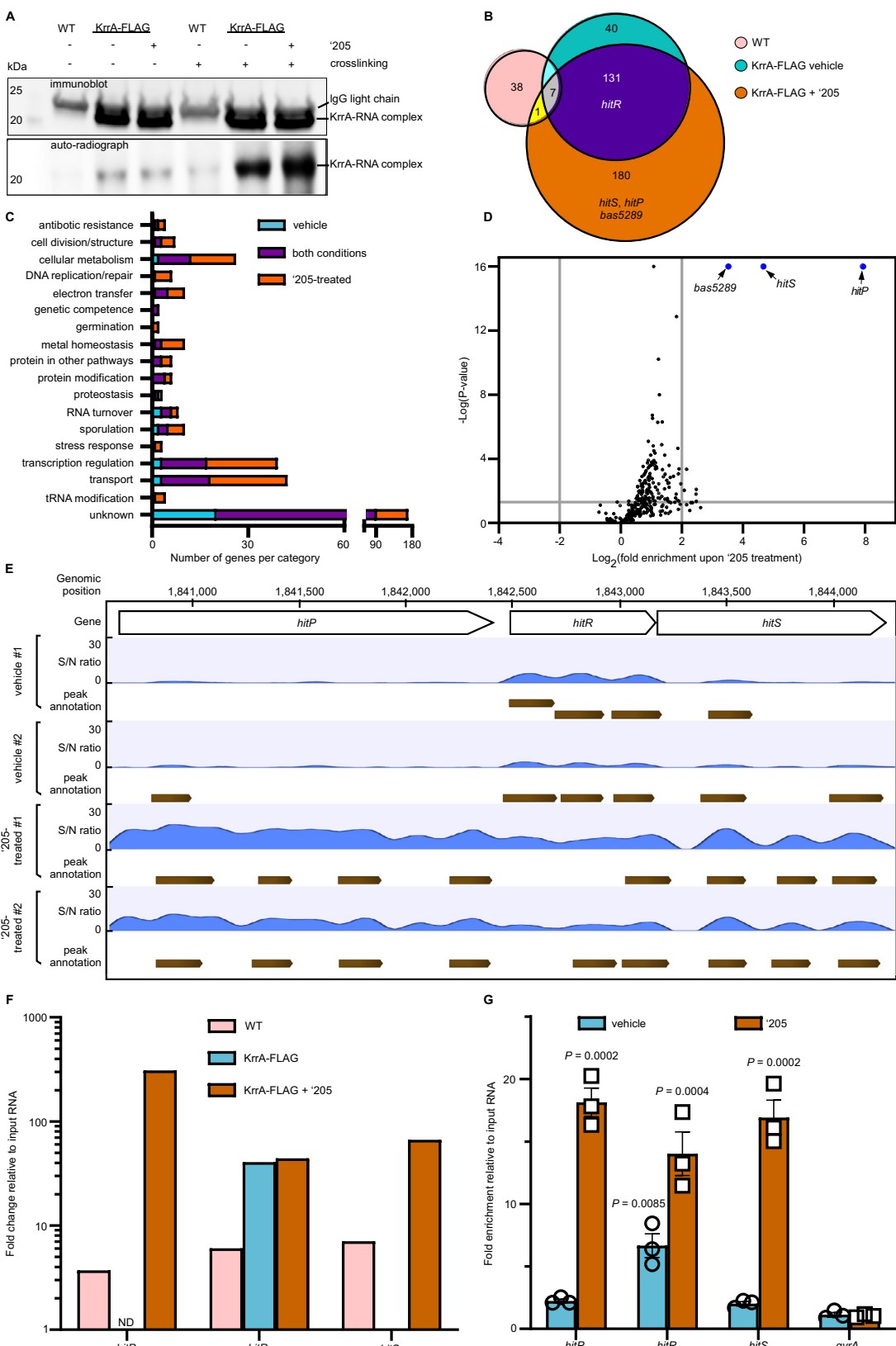

by KrrA under both vehicle- and HitRS-activating conditions, while ~33% or 15% were affected by KrrA only following either vehicle or HitRS activation, respectively (Fig. 5C). Besides genetic competence and transcriptional regulation, the KrrA-regulated genes are involved in many other biological processes, including cellular metabolism, respiration, transport, cofactor synthesis,

proteostasis, metal homeostasis, stress response, germination, and sporulation (Fig. 5D).

We next sought to determine if KrrA-dependent effects on transcription and transcript abundance impact the biology of *B. anthracis*. Sporulation is a sophisticated developmental adaptation to nutrient starvation, which enables *B. anthracis* to survive

**Fig. 3 KrrA directly binds to many RNA targets in vivo, including *hitPRS*. A** To evaluate the RNA-binding ability of KrrA, a CLIP assay (UV crosslinkingg immunoprecipitation) was carried out in WT and Δ*krrA* pOS1.P$_{lgt}$*krrA*-3xFLAG strains in the presence or absence of '205 treatment with or without UV crosslinkingg. The immunoprecipitated KrrA-3xFLAG protein was detected by Western blot using an anti-FLAG antibody (top panel). This serves as a control to evaluate the immunoprecipitation efficiency. The immunoprecipitated and radioactively labeled RNA-KrrA complexes were detected by phosphorimaging (bottom panel). The experiments were conducted at least three times and representative images are shown. Source data are provided as a Source Data file. **B** Overview of KrrA-RNA-binding profiles revealed by formaldehyde cross-linking RNA immunoprecipitation coupled with Illumina sequencing (fRIP-seq) under vehicle or '205-treated conditions. **C** Genes identified in the RIP-seq analysis with significantly increased enrichment grouped into predicted functional categories. **D** Volcano plot showing KrrA-binding RNA transcripts enriched following '205 treatment compared to vehicle. Transcripts that were most significantly enriched upon '205 treatment are highlighted in blue. The log$_2$(fold change) cutoff is ≥2 and the *P* value cutoff is <0.05, both of which are indicated by gray lines. **E** A zoom-in example of KrrA binding to the *hitPRS* transcripts identified by fRIP-seq. Two biological replicates were included for each condition. S/N denotes the signal-to-noise ratio for peak calling. **F** KrrA specifically binds to *hitR* transcript under vehicle conditions and transcripts of the entire *hitPRS* operon upon '205 treatment. ND not detectable. **G** KrrA binding to the transcripts of *hitPRS* was confirmed by fRIP-qPCR under vehicle or '205-treated conditions. The gene *gyrA* was used as a control. The data are expressed as the mean ± SD ($n = 3$). Significant differences are determined by two-tailed *t* tests. Source data are provided as a Source Data file.

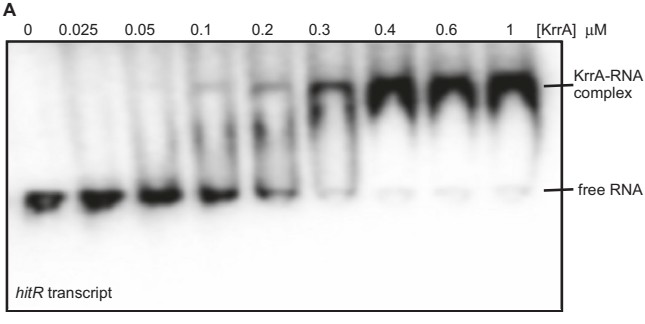

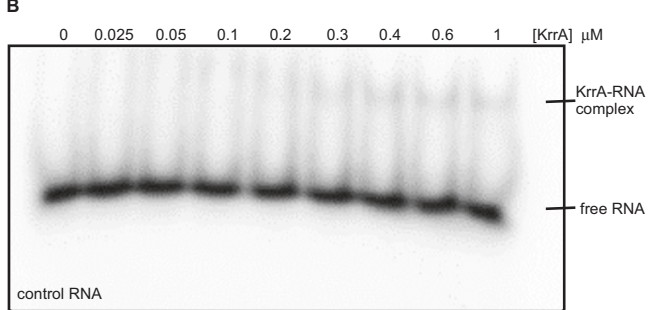

**Fig. 4 KrrA binds to *hitR* transcript in vitro. A, B** To examine whether KrrA binds to *hitR* transcript in vitro, an electrophoretic mobility gel shift assay was carried out using in vitro-transcribed and radioactively labeled *hitR* transcript (**A**) and control RNA (**B**) in the presence of increasing concentration of KrrA protein as indicated. The experiments were conducted at least three times and representative images are shown.

dormant for years. When spores sense germinants and nutrients, they germinate and replicate in the vertebrate host, resulting in the disease anthrax[27]. RNA-seq analysis revealed that transcription of 28 genes involved in various stages of sporulation from onset to spore maturation, and 11 genes important for spore germination were significantly elevated in Δ*krrA* compared to WT (Fig. 5D, E). Two-thirds of these genes (26 out of 39) were affected under both vehicle- and HitRS-activating conditions (Fig. 5D, E and Supplementary Data 2). Notably, following KrrA immunoprecipitation, eight transcripts coding for sporulation proteins (*spoIID*, *spoIIE*, *spoVR*, *spoVM*, *spoVID*, *BAS3655*, *yqfD*, and *yabG*) and two transcripts coding for germination proteins (*gerPA* and *gerPF*) were significantly enriched (Fig. 3C and Supplementary Data 1). Among these ten transcripts, three of them (*spoVR*, *spoVID*, and *BAS3655*) were significantly elevated in Δ*krrA* compared to WT (Fig. 5D, E and Supplementary Data 2). To further evaluate the effects of KrrA function on these processes, we examined the efficiency of both sporulation and

germination in WT and Δ*krrA*. Deletion of *krrA* resulted in significantly higher sporulation efficiency: ~60% in WT compared to ~90% in Δ*krrA*. Similarly, Δ*krrA* showed over 3-fold higher germination efficiency compared to WT (Fig. 5D, inset). Altogether, these data demonstrate the impact of KrrA-mediated RNA regulation on two of the most critical processes in the *B. anthracis* life cycle: sporulation and germination.

**KrrA modulates gene transcription through coregulatory gene networks.** Many transcripts were enriched by KrrA immunoprecipitation only following '205 treatment (Fig. 3) and expression of some genes was altered by KrrA-mediated RNA regulation only under '205-treated conditions (Fig. 5). To reveal the effects of '205 on KrrA function, we next sought to define the transcriptomic changes in response to '205 in *B. anthracis* WT. As expected, *hitP*, *hitR*, and *hitS* showed the highest induction following '205 treatment, with fold changes of 97, 39, and 40, respectively, relative to vehicle (Fig. 6A and Supplementary Data 3)[6]. In addition to the *hit* operon, 108 other genes were significantly upregulated while 170 genes were downregulated following '205 treatment (Fig. 6A, B and Supplementary Data 3). These genes are involved in a variety of biological processes such as transport, DNA replication, transcriptional regulation, stress response, cellular metabolism, TCA cycle, and fermentation (Fig. 6B). Among the nine genes exhibiting the strongest reduction in expression following '205 treatment, two are unique to gluconeogenesis (*pckA* and *gapB*) and three are dual-function small regulatory RNAs that encode SR1P peptides modulating mRNA degradation in *B. subtilis*[28] (Fig. 6C and Supplementary Data 3). In addition, 24 genes involved in gluconeogenesis, TCA cycle, and butanoate fermentation were significantly downregulated while *gapA*, encoding the glycolytic enzyme glyceraldehyde 3-phosphate dehydrogenase, *ldh*, responsible for fermentation of pyruvate to lactate, and *adhE*, involved in ethanol fermentation, were notably upregulated following '205 treatment (Fig. 6C). To further examine the effects of '205 on ethanol fermentation, the ethanol content in the spent media of *B. anthracis* WT culture was quantified. Following treatment of '205, the ethanol content was approximately four times higher relative to vehicle (Fig. 6D). These data indicate that besides HitRS activation, '205 extensively impacts bacterial physiology and diverts cellular metabolism from aerobic respiration to fermentation.

In addition, '205 treatment showed no significant effects on *krrA* gene transcription (Supplementary Data 3). Among 281 genes with differential expression following '205 treatment and 774 genes altered by KrrA function, only 20 genes were affected by both '205 treatment and KrrA-mediated RNA regulation. These genes encode HitR, a TCS pair (BAS2999-3000), transcription regulators (CggR and BAS1801), multiple

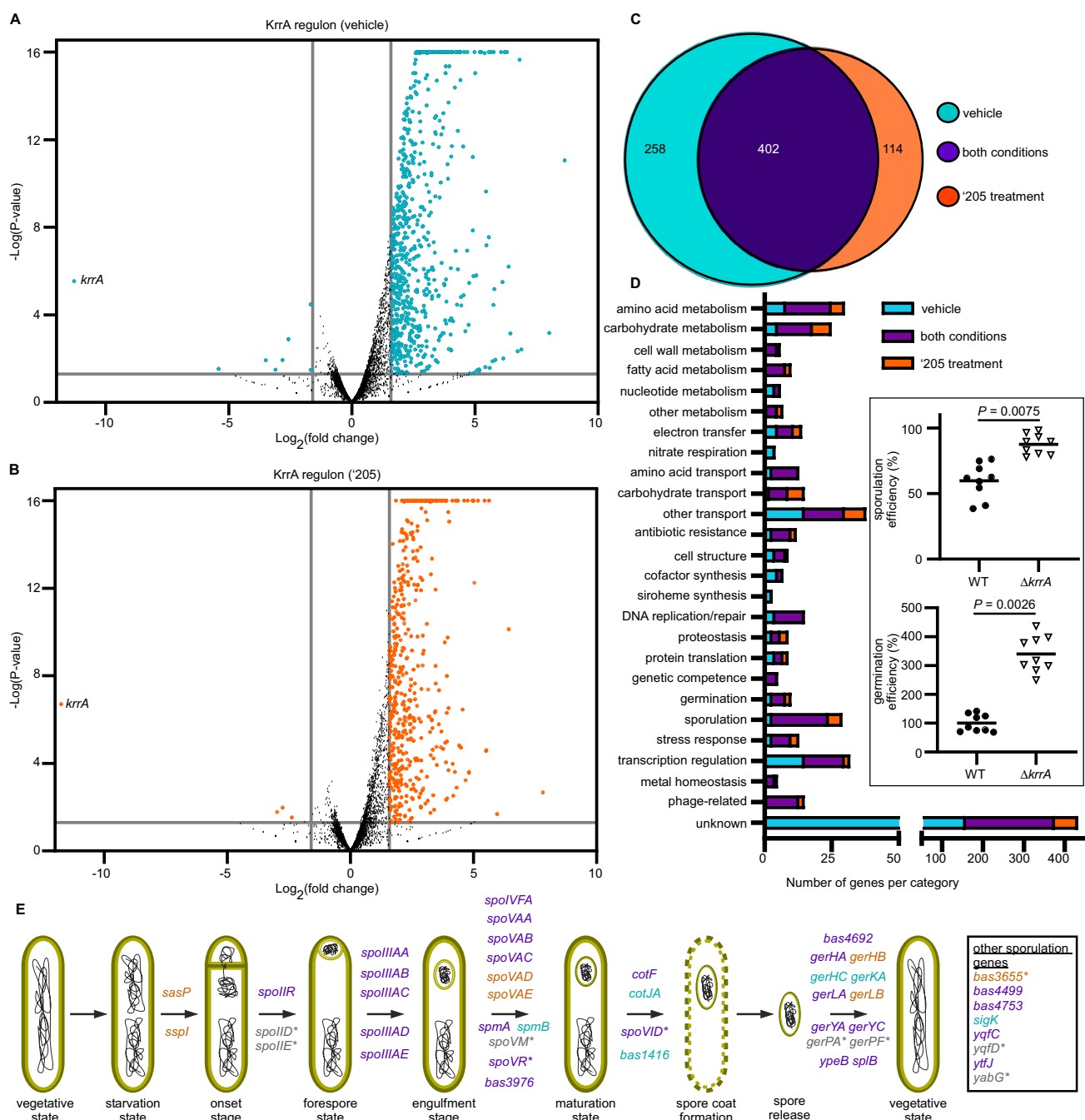

**Fig. 5 KrrA-mediated RNA regulation negatively affects the abundance of numerous transcripts.** Volcano plot showing transcriptomic comparison between *B. anthracis* Δ*krrA* and WT following vehicle (**A**) or 50 μM '205 treatment (**B**). Genes with significantly different expressions are highlighted in cyan (**A**) or orange (**B**). The fold change cutoff is ≥3 and the *P* value cutoff is <0.05, both of which are indicated by gray lines. **C** Overview of KrrA targets revealed by RNA-seq under vehicle or '205-treated conditions. **D** Genes identified in the RNA-seq analysis with significantly differential expression grouped into predicted functional categories. Inset: Efficiency of sporulation and germination was further examined in *B. anthracis* WT and Δ*krrA*. Data shown are three independent experiments with three replicates each time (mean ± SEM; *n* = 9). Significant differences are determined by two-tailed *t* tests. **E** The key stages in the cycle of sporulation and germination showing all genes with significantly elevated expression identified from RNA-seq under vehicle- (cyan), '205-treated (orange), or both conditions (purple). Other genes that have not been characterized for any specific stage are listed in the box. Additional genes with minor changes in expression but identified from RIP-seq are shown in gray. The genes enriched by KrrA immunoprecipitation (RIP-seq) are denoted by asterisk.

transporters, exodeoxyribonuclease III (Xth) involved in DNA repair, bacterioferritin (BAS0929), cytochrome C oxidase (CtaF), and proteins with unknown function (Supplementary Data 3). These data suggest that '205 does not play a major role in KrrA function.

Transcription and mRNA degradation are linked processes and tight regulation of mRNA stability is critical for the control of gene expression. To define the global effects of KrrA function on both transcription and RNA turnover, cross-comparisons of RIP-seq and RNA-seq datasets were carried out. The majority

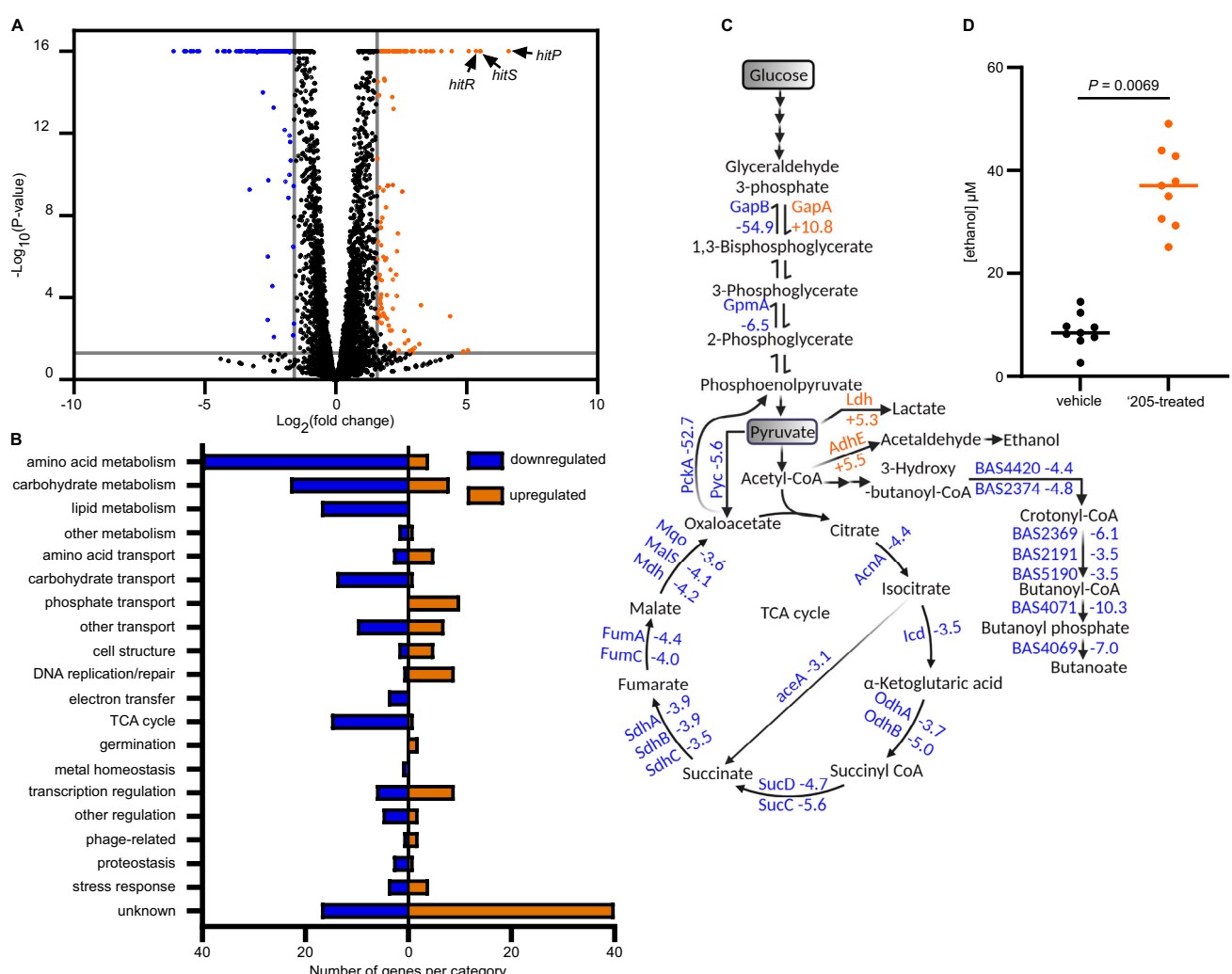

**Fig. 6 The HitRS activator '205 diverts cellular metabolism from aerobic respiration to fermentation. A** Volcano plot showing transcriptomic comparison between '205 (50 μM) and vehicle treatments of *B. anthracis* WT. Genes significantly upregulated or downregulated following '205 treatment are highlighted in orange or blue, respectively. The fold change cutoff is ≥3 and the *P* value cutoff is <0.05, both of which are indicated by gray lines. **B** Genes identified in the RNA-seq analysis with significantly differential expression grouped into predicted functional categories. **C** A subset of notable metabolic pathways that showed significantly different expression following '205 treatment: glycolysis, gluconeogenesis, and TCA cycle. Products of genes with increased expression are labeled in orange while the ones with decreased expression are labeled in blue. The fold change of each specific gene is indicted: +, upregulation; −, downregulation. **D** Ethanol concentration was quantified in the spent media of WT cell cultures in the absence or presence of 50 μM '205. '205 drives glucose metabolism towards pyruvate fermentation, resulting in higher level of ethanol compared to vehicle control. Data shown are three independent experiments with three replicates each time (mean ± SEM; *n* = 9). Significant differences are determined by two-tailed *t* tests.

of the KrrA-interacting RNA targets (286 out of 359) showed minor changes in transcript abundance (Fig. 7), suggesting that some of these RNAs may not be authentic KrrA targets or that KrrA function does not affect their gene expression under the growth conditions tested. Nevertheless, KrrA regulation altered the mRNA levels of 73 targets including *hitR*. The transcript abundance of *hitR* in the RNA-seq experiment was approximately threefold higher in Δ*krrA* relative to WT under vehicle-treated conditions (Fig. 7 and Supplementary Data 4). Approximately 50% of the transcripts (34 out of 73) were enriched by KrrA binding under both vehicle and '205-treated conditions while many of them (33 out of 73) were only enriched following '205 treatment. Moreover, seven other transcription regulators including *sigI*, which encodes the sigma factor involved in heat shock stress[29], showed significantly higher transcript levels in Δ*krrA* compared to WT (Fig. 7 and Supplementary Data 4). The majority of the transcripts

with differential expression (701 out of 774) showed no significant enrichment upon KrrA binding in the RIP-seq dataset (Fig. 7 and Supplementary Data 4). These data indicate that KrrA has both direct and indirect roles in the regulation of gene transcription.

In addition to transcriptional regulation, the 73 targets of KrrA are also involved in genetic competence, sporulation, RNA turnover, DNA repair, transport, and cellular metabolism (Fig. 7 and Supplementary Data 4). Specifically, KrrA directly or indirectly regulates the expression of over 150 genes encoding enzymes of cellular metabolism and transporters of carbohydrates and other nutrients (Fig. 7 and Supplementary Data 4), indicative of a major role of KrrA in these two processes. Altogether, these data demonstrate that KrrA functions as a pleiotropic post-transcriptional modulator and KrrA-mediated RNA regulation extensively controls gene transcription through coregulatory networks.

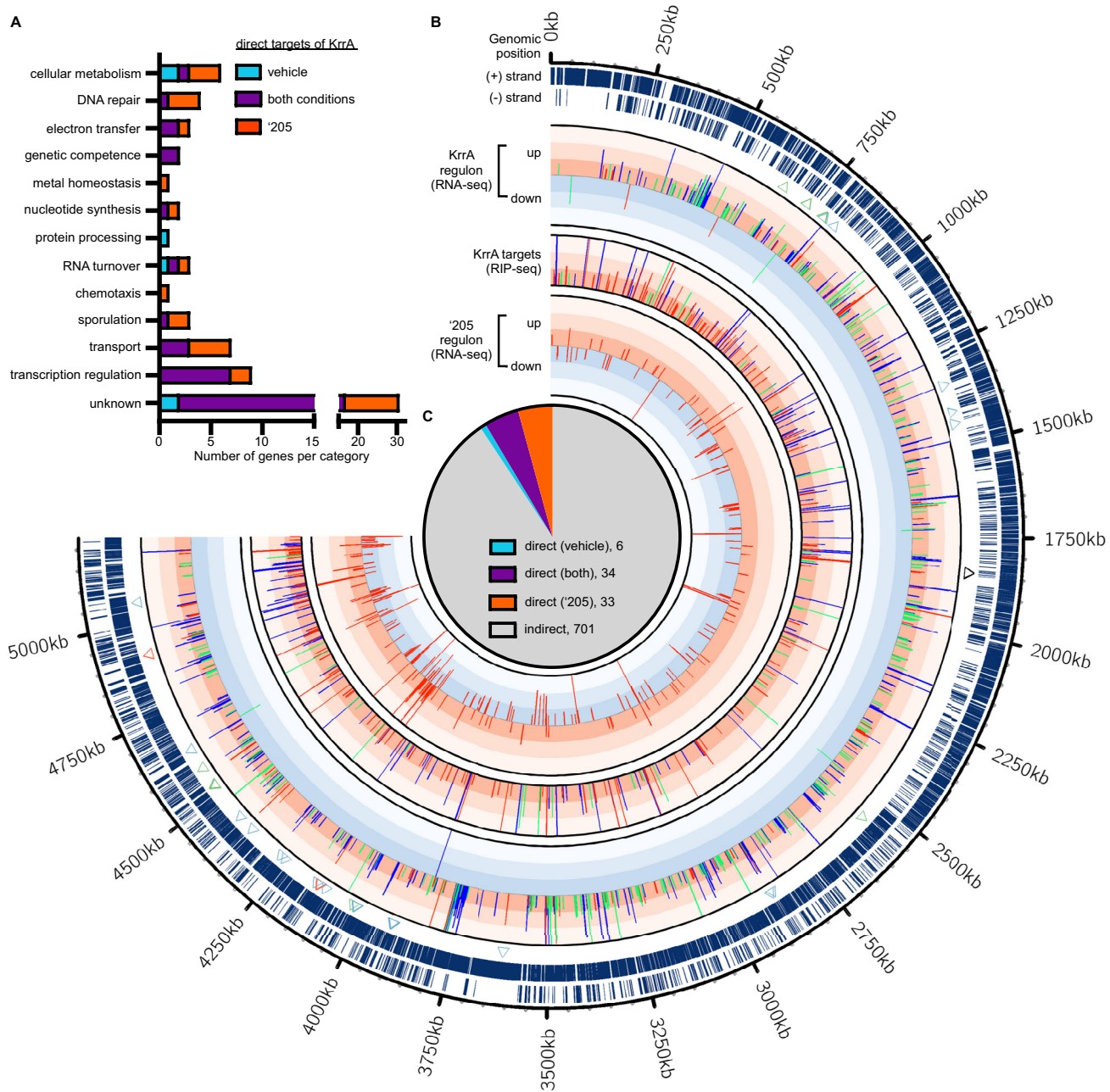

**Fig. 7 Combined fRIP-seq and RNA-seq datasets uncover the broad effects in gene expression and coregulatory gene networks modulated by KrrA function.** **A** The transcript abundance of 73 direct targets was significantly affected by KrrA regulation and these direct RNA targets are grouped into various functional categories. **B** Combined fRIP-seq and RNA-seq data showing global changes modulated by KrrA function. Each bar represents a gene with significantly differential abundance. Each level of color gradation represents a fivefold change. Genes with ≥15-fold change are represented with bars that hit the maximum limit. The bars in different colors show genes with differential abundance under vehicle (green), '205-treated (red), or both conditions (blue). The triangles indicate the KrrA-regulated transcripts that are involved in TCS signaling (black), competence (red), sporulation (green), and germination (blue). **C** Overview of KrrA targets revealed by comparing the datasets of fRIP-seq and RNA-seq.

## Discussion

The molecular basis by which bacterial TCSs drive signal transduction is well understood, however, limited information is available regarding accessory proteins involved in the regulation of these signaling systems. In this study, we employed an unbiased genetic selection strategy that identified a TCS modulator KrrA that is critical for regulating HitRS signal transduction, and further dissected a regulatory mechanism of TCS signaling that involves titrating transcript abundance. We demonstrated that KrrA functions as an RNA-binding protein, binds numerous RNA transcripts in vivo, and plays a post-

transcriptional regulatory role in many cellular processes including TCS signaling. TCS genes are generally transcribed at some basal level to enable microbes to sense and respond to rapidly changing environments in a timely manner. In *B. anthracis*, KrrA specifically binds to the *hitR* transcripts, facilitates mRNA turnover, and executes tight control over HitRS signal transduction that detects cell envelope disruptions.

The importance of bacterial post-transcriptional regulation by RBPs has been increasingly appreciated in Gram-positive bacteria in recent years, however, the mechanisms of these regulatory networks remain understudied[25,30]. A recent study demonstrated

that transcription and translation are functionally uncoupled in the Gram-positive model organism *B. subtilis*[31], suggesting that in Gram-positive bacteria mRNA quality control through RBPs may play a major role in gene regulation rather than translational coupling. Data presented here demonstrate that KrrA is such an RBP. Generally, RBPs bind specific RNAs and modulate RNA degradation, transcription termination, or translation initiation of their targets using various regulatory strategies[32–35]. Post-transcriptional control of gene expression provides at least two major fitness advantages to bacteria growing in constantly changing environments: (i) it enables a rapid and robust response to alter expression at both gene and protein levels; (ii) it allows cells to fine-tune protein expression by segregating transcription and translation[32–34]. The latter is specifically important for functionally related genes that are located in the same operon and under the same transcriptional regulation. Different levels of these proteins may be required for cellular functions. For example, TCS gene pairs are often organized in the same operon and co-transcribed, but RR proteins are typically more abundant compared to their cognate HKs, which is beneficial to bacterial fitness by allowing a robust response to specific stimuli[36]. Differential expression of co-transcribed genes is effectively orchestrated by proportional protein synthesis[36] and may also be modulated by differential mRNA degradation. This study illustrates that an RBP interacts with a RR transcript and facilitates mRNA turnover of the RR to precisely tune the cellular response to ensure specificity of TCS signaling in a bacterial pathogen. This regulation is likely a common strategy to modulate differential expression within transcription operons including TCS pairs. Consistent with this, CsrA, a widely distributed post-transcriptional regulator, is required for the proper expression of the RR UvrY[37]. CsrA also interacts with many other TCS transcripts and alters their gene expression although its role in activation of these TCSs has yet to be defined[38].

A detailed global view of the diverse effects of KrrA-mediated RNA regulation on gene expression was shown through integrated transcriptomics approaches. These data demonstrate that KrrA functions as a pleiotropic post-transcriptional modulator, reshapes the transcriptome in response to constantly changing environments, and is engaged in regulating important bacterial processes such as genetic competence, sporulation, and germination (Supplementary Fig. 2 and Fig. 5). All three of these processes are energy-consuming modules orchestrated by complex cellular networks. For example, in *B. subtilis*, a naturally competent model organism, the competence module is choreographed by a ComK-MecA-ComS circuit that functions as a bistable system and the TCS ComAP that detects the external quorum-sensing signal ComX[39]. Expression of the master competence regulator ComK is tightly controlled at three different levels: transcriptional regulation by at least six different regulators including ComK itself[40,41], protein degradation by the MecA/ClpP/ClpC complex[42], and mRNA turnover likely by Kre[24]. *B. anthracis* is not naturally competent but harbors similar genetic exchange systems that enable transduction, conjugation, and transformation[43]. KrrA interacts with *comFA* and *comFC* transcripts (Fig. 3), which are required for DNA uptake during transformation, tunes their gene expression (Fig. 5), and imposes a post-transcriptional control in addition to transcription regulation by ComK. This robust post-transcriptional regulation in turn drastically affects DNA transformation efficiency (Supplementary Fig. 2A, B). Furthermore, these data along with others[24] suggest that these *Bacillus* RBPs play a common role in genetic competence through post-transcriptional regulation although the Kre-interacting RNA targets await characterization in *B. subtilis*.

Heretofore a number of bacterial RBPs have been identified, however, only three of which have been intensively studied as global post-transcriptional regulators: Hfq, ProQ, and CsrA[44,45]. Hfq and ProQ facilitate interactions between small regulatory RNAs (sRNAs) and target mRNA transcripts and assist the recruitment of the RNA degradosome complex. CsrA is antagonized by two sRNAs, CsrB and CsrC, which control CsrA protein level and modulate CsrA binding to mRNA targets. In addition, these three RBPs can directly bind to their RNA targets and mediate RNA turnover in an sRNA-independent manner[44,45]. Furthermore, each RBP possesses different RNA-binding motifs: a tripartite (A-R-E) motif for Hfq[46], a N-terminal FinO-domain for ProQ[47], and a KH domain with a GxxG core sequence for CsrA[48]. KrrA is a newly identified RBP without a known RNA-binding motif. Notably, the transcript exhibiting the highest enrichment upon KrrA immunoprecipitation encodes an 86 bp pseudogene *BAS_RS13510*, with fold enrichment of 2029 or 3402 under vehicle or HitRS-activating conditions, respectively (Supplementary Table 4). This pseudogene is only present in the three pathogenic Bacilli: *B. anthracis*, *B. cereus*, and *B. thuringiensis*. It is truncated from the 3'-end of *rnaY* that encodes a HD-domain containing phosphohydrolase[49]. Pseudogenes are frequently assigned roles as sRNAs that are important for RNA regulation[50]. In this regard, 19 additional intergenic regions were found to be specifically enriched upon KrrA binding, with sequencing reads of ~50 bp on average and fold of enrichment up to 5800 (Supplementary Table 4). These data suggest that these enrichment sites along with the pseudogene *BAS_RS13510* may encode sRNAs that interact with KrrA during RNA degradation. The physiological roles of these putative sRNAs are currently under investigation. Collectively, KrrA functions as a versatile post-transcriptional modulator. This protein specifically binds to target mRNAs and putative sRNA regulators and broadly impacts the expression of functionally coordinated sets of genes both directly and indirectly through cellular networks. In addition, KrrA coordinates with multiple signaling networks to enable *B. anthracis* to precisely direct cellular responses in times of stressful conditions. These findings provide new insights into regulatory mechanisms of TCS signaling and expand our understanding of bacterial post-transcriptional regulation.

## Methods

**Bacterial strains and growth conditions**. All strains and plasmids used in this study are listed in Supplementary Table 1. Cells were grown in LB with shaking at 180 RPM or on solid LB agar plates with the appropriate antibiotic selection at 37 °C. The concentrations of antibiotics used are carbenicillin (carb, 50 μg ml$^{-1}$), chloramphenicol (cam, 30 μg ml$^{-1}$), kanamycin (kan, 40 μg ml$^{-1}$), and erythromycin (erm, 20 μg ml$^{-1}$). Stocks of the compound '205 (50 mM) were made in DMSO and stored at −20 °C.

**DNA manipulation and strain construction**. The generation of a plasmid for deletion of *krrA* was performed by inserting flanking sequences in the mutagenesis plasmid pLM4[51]. Briefly, flanking sequences were amplified using a distal primer containing a restriction enzyme site found in pLM4 (XmaI or SacI) and a proximal primer containing a short overlapping sequence. PCR-amplified DNA was fused using PCR sequence overlap extension (PCR-SOE)[52] and inserted between the XmaI and SacI sites in pLM4 to generate the plasmid pLM4-*krrA*. Genetic manipulation in *B. anthracis* was conducted using pLM4-*krrA* to generate *B. anthracis* Δ*krrA*. To make a recombinant KrrA expression construct, the coding sequence of *krrA* was amplified by PCR from the *B. anthracis* genome and cloned into pET15b expression vector to generate a 6x-histidine-tagged protein at the N-terminus. The recombinant plasmid was transformed into *E. coli* DH5α, confirmed by Sanger sequencing, then subsequently transformed into *E. coli* BL21 (DE3) pREL for protein expression and purification. All strains used in this study were verified by PCR using primers listed in Supplementary Table 2.

**Genetic selection**. To isolate spontaneous erythromycin-resistant suppressor mutants that constitutively activate HitRS signaling, we utilized strain *bas3009::hit-ermC* (referred to herein as P$_{hit}$*ermC*)[9]. The strain P$_{hit}$*ermC* was streaked onto an LB agar plate and single colonies were grown in 5 ml LB for 18 h at 30 °C with vigorous shaking. Serial dilutions of the bacterial cultures were made (typically 1:3, 1:10, 1:30, and 1:100). Each dilution (100 μl) was plated onto LB containing 5, 10, or 20 μg ml$^{-1}$ erythromycin. Plates were incubated at 30 °C for up to 3 days.

Spontaneously arising colonies exhibiting erythromycin resistance were isolated, streaked for single colonies on fresh LB agar plates, and saved for further analysis.

**Whole-genome sequencing.** Genomic DNA (gDNA) was extracted from the erythromycin-resistant suppressor mutants using the Qiagen DNeasy Blood and Tissue kit according to the manufacturer's instructions. Purified gDNA was sequenced by Genewiz using the Illumina NextSeq 550 platform with 150 Mbs coverage. Sequencing reads were trimmed and mapped to the genome of *B. anthracis* Sterne (T00184), and all mapped reads were subjected to variant detection using CLC Genomics Workbench v20.0.1 with default settings.

**Growth curves.** Cells were grown overnight in LB medium at 30 °C, subcultured at a 1:100 ratio into fresh LB medium and grown for 6 h at 37 °C with vigorous shaking. Cell density ($OD_{600}$) was then monitored every 30 min for 24 h at 37 °C with continuous shaking using a BioTek Epoch2 spectrophotometer. Experiments were conducted at least three times with three biological replicates each time. Data shown are averages of three biological replicates (mean ± STD).

**DNA transformation efficiency assay.** In *B. subtilis*, transformation efficiency was quantified by transforming competent cultures with genomic DNA carrying a kanamycin resistance marker. Strains were grown to stationary phase for 7 h in modified competence medium (MC medium: 100 mM potassium phosphate buffer pH 7.0, 3 mM trisodium citrate, 3 mM $MgSO_4$, 2% glucose, 0.2% potassium glutamate, 0.1% casein hydrolysate, 22 µg ml$^{-1}$ ferric ammonium citrate, and 50 µg ml$^{-1}$ tryptophan). Chromosomal DNA (1 µg) was added to 1 ml of the culture and incubated for 1 h at 37 °C with rigorous agitation. Cell cultures were serially diluted for CFU counts on LB agar plates without or with supplementation of kanamycin. Transformation efficiencies were calculated by dividing the number of transformants by the viable CFU count of each culture. IPTG (1 mM) was added to the MC medium and kanamycin plates to induce the expression of *B. anthracis krrA*.

In *B. anthracis*, transformation efficiency was quantified using electroporation of either WT or Δ*krrA* electrocompetent cells with plasmid DNA pOS1.P*hit*xylE harboring a chloramphenicol resistant marker. Plasmid DNA (2 µg) was added to 50 µl of electrocompetent cells and incubated on ice for 5 min. The mixture was then subjected to electroporation with setting as follows, time: 1 ms; resistance: 100 Ω; and capacitance: 10 µF, amended with 1 ml BHI broth containing 0.5 M D-sorbitol, incubated for 5 h at 30 °C with rigorous agitation. Cell cultures were then serially diluted for CFU counts on LB agar plates without or with supplementation of chloramphenicol. Transformation efficiencies were calculated by dividing the number of transformants by the viable CFU count of each culture.

**RNA extraction and quantitative PCR (qPCR).** Cells were grown at 30 °C in LB medium overnight and subcultured at a 1:100 ratio into fresh LB medium. After 6 h of growth at 37 °C in the presence or absence of TCS activator as specified, aliquots of 1 ml of cell culture were harvested by centrifugation. Total RNA was extracted using a RNeasy Mini Kit following the manufacturer's instructions (Qiagen Sciences, Germantown, MD), and treated with Turbo-DNA-free DNase (Ambion). The DNase was removed using DNase removal reagents (Ambion). RNA samples were quantified using a NanoDrop spectrophotometer. Total RNA (200 ng) from each sample was subjected to cDNA synthesis using high-capacity cDNA reverse transcription kits (Applied Biosystems, Foster City, CA). Quantitative PCR (qPCR) was then conducted using iQ SYBR green supermix (Bio-Rad) on a CFX96 qPCR cycler (Bio-Rad). The *B. anthracis* housekeeping gene 16 S rRNA was used as an internal control.

**XylE reporter assay.** To examine the effects of *krrA* deletion on TCS activation, XylE reporter constructs were generated to evaluate the promoter activity of either P*hit* or P*eds*. Cells were grown at 30 °C in LB medium overnight and subcultured at a 1:100 ratio into fresh LB medium. After 6 h of growth at 37 °C in the presence or absence of TCS activator as specified, the abundance of the XylE enzyme (catechol 2, 3-dioxygenase) present in *B. anthracis* cell lysates was quantified by measuring the rate at which catechol was converted to 2-hydroxymuconic acid using a spectrophotometer.

**Northern blot.** Total RNA was extracted using a RNeasy Mini Kit following the manufacturer's instructions (Qiagen Sciences, Germantown, MD), and treated with Turbo-DNA-free DNase (Ambion). The DNase was removed using DNase removal reagents (Ambion). RNA samples were quantified using a NanoDrop spectrophotometer. RNA (15 µg) was mixed with loading buffer (formamide, bromophenol blue, xylene cyanol, EDTA), denatured at 90 °C for 2 min, loaded on a 1% bleach agarose gel, and run at 100 W for ~2 h. RNA was transferred onto nylon membrane in 20× SSC buffer at room temperature overnight followed by cross-linking two times at 254 nm. The membrane was incubated in DIG easy hybridization buffer at 42 °C for 1 h. Four antisense-oligo probes against *hitPRS* (Supplementary Table 2) were pooled together and labeled at 5'-ends with [γ-³²P]-ATP using T4 polynucleotide kinase. After labeling, G10 columns (NucAway spin columns, Invitrogen) were used to remove the unincorporated (γ-³²P) ATP. The labeled probes were added to the hybridization buffer and incubated at 42 °C

overnight. One labeled probe against 16 S rRNA (Supplementary Table 2) served as a loading control. The membrane was washed 3× in wash buffer (0.5× SSC, 0.1% SDS) followed by exposure to a phosphorimager screen overnight, and the radioactive signals were detected using a phosphor image analyzer (Typhoon FLA 7000).

**mRNA stability assay.** Cells were grown at 30 °C in LB medium overnight and subcultured at a 1:100 ratio into fresh LB medium. After 5 h of growth at 37 °C, aliquots of 4 ml of cell culture were mixed with 4 ml of ice-cold acetone/EtOH (1:1 ratio) and harvested by centrifugation. These served as $T_0$ samples. Rifampicin was added to the remaining cell cultures to a final concentration of 150 µg ml$^{-1}$. Aliquots of 4 ml of cell culture were harvested as described above at time intervals following rifampicin treatment. Total RNA was extracted using RNeasy Mini Kit following the manufacturer's instructions (Qiagen Sciences, Germantown, MD) and treated with Turbo-DNA-free DNase (Ambion). The DNase was removed using DNase removal reagents (Ambion). RNA samples were quantified using a NanoDrop spectrophotometer. Total RNA (200 ng) from each sample was subjected to cDNA synthesis using high-capacity cDNA reverse transcription kits (Applied Biosystems, Foster City, CA). Quantitative PCR (qPCR) was then conducted using iQ SYBR green supermix (Bio-Rad) on a CFX96 qPCR cycler (Bio-Rad). The *B. anthracis* housekeeping genes 16 s rRNA and *gyrA* were used as controls.

**Protein expression and purification.** KrrA was expressed in *E. coli* BL21 (DE3) pREL. Cells were grown overnight in LB medium at 30 °C, subcultured at a 1:100 ratio to fresh LB medium, and grown for ~4 h at 30 °C with vigorous shaking ($OD_{600}$ ~0.6–0.8). Cell cultures were then switched to incubation at 18 °C and IPTG (0.5 mM) was added to induce protein expression. After 18 h of incubation at 18 °C, cells were harvested by centrifugation. Cells were lysed by sonication and the lysate was centrifuged at $20,000 \times g$ for 45 min at 4 °C to remove the debris. The subsequently clarified lysate was loaded onto an NTA-Ni column and protein was eluted sequentially with an increasing concentration of imidazole. The fractions containing high-purity proteins were pooled and subjected to dialysis for buffer exchange using storage buffer containing 50 mM Tris (pH 8), 300 mM NaCl, 0.5 mM EDTA, 1 mM DTT, and 20% glycerol, and concentrated using Amicon concentrators. Purified proteins were subjected to SDS-PAGE for quality control analysis and quantified using absorbance at 280 nm on a NanoDrop spectrophotometer. Aliquots of purified proteins were flash-frozen in liquid nitrogen and stored at −80 °C.

**Western blotting to evaluate immunoprecipitation efficiency and HitS protein levels.** Following immunoprecipitation (Fig. 3A, G), the KrrA-3xFLAG fusion protein was eluted from the α-FLAG M2 magnetic agarose beads (Sigma, Cat# M8823), applied to SDS-PAGE electrophoresis. For experiments performed in Supplementary Fig. 4, cells were sonicated in lysis buffer (50 mM Tris [pH 7.5], 150 mM NaCl) amended with protease inhibitor, and cell lysates were subjected to ultracentrifugation at $150,000 \times g$ for 1 h to separate the membrane fractions, which were resolved by SDS-PAGE. Following electrophoresis, the protein samples were transferred to a 0.45-µm-pore-size nitrocellulose membrane using a Trans-Blot Turbo system (Bio-Rad, Hercules, CA) followed by immunoblotting. The primary antibodies used were monoclonal α-FLAG antibody (Sigma-Aldrich, cat # F1804, 1:1000 dilution), α-myc (Abcam, cat# ab9132, 1:1000 dilution), and α-SrtA rabbit serum (1:2000 dilution)[53]. The blots were scanned using an Odyssey Imager. The FLAG-tagged KrrA or Myc-tagged HitS has a molecular mass of ~21 or 44 kDa, respectively, in accordance with the signals observed in the blots. The blots shown in Supplementary Fig. 4 are representative images of three independent experiments. Densitometry analysis of band intensity for HitS was performed using GelQuantNET. Uncropped scans of the most important blots are included in Source Data.

**DNA and RNA nuclease activity assays.** For the DNA nuclease activity assay, pET15b plasmid DNA (500 ng per reaction) harboring an EcoRI restriction site was incubated with either EcoRI (positive control) or varied concentrations of KrrA ranging from 0 to 1 µM in either CutSmart buffer or a phosphate buffer (20 mM phosphate pH 7.5, 160 mM NaCl, 40 mM KCl, and 4% (vol/vol) glycerol) at 37 °C for 1 h. The reaction products for pET15b plasmid DNA were detected by electrophoresis in 1% agarose gels with ethidium bromide staining.

For the RNA nuclease activity assay, total RNA (2 µg per reaction) extracted from *B. anthracis* WT was incubated with either RNase A (positive control) or varied concentrations of KrrA ranging from 0 to 10 µM in a phosphate buffer (20 mM phosphate pH 7.5, 160 mM NaCl, 40 mM KCl, and 4% (vol/vol) glycerol) at 25 °C for 30 min. The reaction products were detected by electrophoresis in 1.2% formaldehyde agarose gels and visualized by ethidium bromide staining.

**UV cross-linking immunoprecipitation (CLIP).** *B. anthracis* cells (WT and WT pOS1.P*lgt*.*krrA*-FLAG) were grown at 30 °C in LB medium overnight and subcultured at a 1:100 ratio into fresh LB medium. After 6 h of growth at 37 °C in the presence or absence of 50 µM '205, cells were harvested by centrifugation, washed with ice-cold PBS, and subjected to UV cross-linking (0.8 J cm$^{-2}$, 254 nm). The

crosslinked cells were resuspended in 0.5 ml buffer A (50 mM Tris pH 7.4, 150 mM NaCl, and 1 mM EDTA) followed by sonication for cell lysis and DNA fragmentation. The supernatant fraction was collected after centrifugation, treated with RNase T1 (1U per µl) for 15 min at room temperature, and incubated with α-FLAG M2 magnetic agarose beads (Sigma, Cat# M8823) on a rotating wheel for 2 h at 4 °C for immunoprecipitation. The bead slurry was recovered using a magnetic stand, washed three times with 1 ml of IP wash buffer (50 mM Tris pH 7.4, 10 mM MgCl₂, and 0.2% Tween-20), treated with RNase T1 (100 U per µl) for 15 min at room temperature, washed three times with 1 ml of high-salt wash buffer (50 mM Tris pH 7.4, 1 M NaCl, and 1 mM EDTA), and washed once in IP wash buffer. One-tenth of the washed beads were saved for immunoblotting and the remaining samples were subjected to radioactive labeling with [γ-³²P]-ATP using the KinaseMax 5′ End-Labeling Kit (ThermoFisher, Cat# AM1520) according to the manufacturer's protocol. The labeled beads were washed twice in NT2 buffer (50 mM Tris pH 7.4, 150 mM NaCl, 10 mM MgCl₂, and 0.0005% NP-40) and heated at 90 °C for 5 min with 40 µl of SDS-loading buffer (50 mM Tris-Cl pH 6.8, 2% (w/v) SDS, 0.1% (w/v) bromophenol blue, 10% (v/v) glycerol, and 5% 2-mercaptoethanol). The resultant supernatants were applied to SDS-PAGE electrophoresis. The gels were dried using a gel dryer, exposed to a phosphor screen overnight, and scanned using a phosphorimager (Typhoon FLA 7000).

**Formaldehyde cross-linking RNA immunoprecipitation coupled with Illumina sequencing (fRIP-seq) and data analysis.** *B. anthracis* cells (WT and ΔkrrA pOS1.P$_{lgt}$.krrA-FLAG) were grown at 30 °C in LB medium overnight and subcultured at a 1:100 ratio into fresh LB medium. After 6 h of growth at 37 °C in the presence or absence of 50 µM '205, cells were harvested by centrifugation, washed with PBS, subjected to cross-linking using formaldehyde (0.75% final) at room temperature for 10 min, then treated with glycine (125 mM final, pH 7.5) at room temperature for 5 min to quench the cross-linking reaction. The cells were subsequently harvested by centrifugation, washed twice with buffer A (50 mM Tris pH 7.4, 150 mM NaCl, and 1 mM EDTA), and resuspended in 0.5 ml buffer A amended with 1 mM PMSF and 2 µl RNase inhibitor followed by sonication for cell lysis and DNA fragmentation. The cell lysates (supernatants) were collected after centrifugation. Aliquots of 1% lysates were diluted with buffer A and kept at −80 °C to serve as the input control (1% of input RNA). The remaining lysates were incubated with α-FLAG M2 magnetic agarose beads (Sigma, Cat# M8823) on a rotating wheel for 2 h at 4 °C for immunoprecipitation. The bead slurry was recovered using a magnetic stand, washed twice with 1 ml of high-salt wash buffer (50 mM Tris pH 7.4, 1 M NaCl, and 1 mM EDTA), then washed once with 1 ml of IP wash buffer (50 mM Tris pH 7.4, 10 mM MgCl₂, and 0.2% Tween-20). Aliquots of 5% beads were saved for immunoblotting to examine the immunoprecipitation efficiency. The remaining beads were subjected to DNAse treatment and the protein–RNA complexes were eluted with 3× FLAG peptide according to the manufacturer's protocol. All samples including 1% input RNA samples were incubated in buffer B (50 mM Tris pH 7.4, 10 mM DTT, 1% SDS, 1 mg ml⁻¹ protease K, and 5 mM EDTA) at 70 °C for 45 min to reverse cross-linking. Enriched RNAs were extracted using Trizol, precipitated using isopropyl alcohol with linear acrylamide, washed twice with 75% ethanol, and quantified using a NanoDrop spectrophotometer.

RNA sequencing was performed with two biological replicates per condition by the Vanderbilt Technologies for Advanced Genomics Core Facility (VANTAGE). The RNAs were assessed using the 2100 Bioanalyzer (Agilent) followed by rRNA depletion and cDNA library preparation using the ScriptSeq complete (bacterial) kit (Illumina/Epicentre) following the manufacturer's protocol. The cDNA libraries were assessed using the 2100 Bioanalyzer (Agilent), quantitated using KAPA library quantification kits (KAPA Biosystems), then subjected to paired-end sequencing on a HiSeq3000 with 150 cycles at VANTAGE.

Bioinformatic analysis was performed using CLC genomic workbench software (version 20.0.1) with default settings. Paired-end sequencing data were imported into CLC Genomics Workbench and trimmed to remove barcodes and adapter sequences. To computationally deplete remaining rRNA reads, the entirety of generated reads was aligned to a sequence list containing all rRNA sequences for the genome of *B. anthracis* Sterne (NC_000964). Unmapped (non-rRNA) reads were collected and utilized for RNA-Seq analysis with default settings. Since the sequencing depth may differ between samples, a per-sample library size normalization was performed using a TMM method (trimmed mean of M values)[54]. Comparisons of gene expression patterns under different conditions were then carried out using the "Differential Expression in Two Groups" function. A shape-based peak caller of the CLC genomic workbench was used to compare experiment alignments and control (input RNA) to identify enriched RIP peaks. The threshold for signal to noise ratio (S/N), which is analogous to RIP-DNA enrichment ratio versus input RNA control, was set as 3 and the P value threshold was set as 0.05.

**Formaldehyde cross-linking RNA immunoprecipitation paired with qPCR (fRIP-qPCR).** *B. anthracis* cells (WT and ΔkrrA pOS1.P$_{lgt}$.krrA-FLAG) were grown at 30 °C in LB medium overnight and subcultured at a 1:100 ratio into fresh LB medium. After 6 h of growth at 37 °C in the presence or absence of 50 µM '205, cells were harvested by centrifugation, washed with PBS, and subjected to cross-linking using formaldehyde (0.75% final) at room temperature for 10 min, then treated with glycine (125 mM final, pH 7.5) at room temperature for 5 min to quench the cross-linking. The cells were subsequently harvested by centrifugation, washed twice with buffer A (50 mM Tris pH 7.4, 150 mM NaCl, and 1 mM EDTA), and resuspended in 0.5 ml buffer A amended with 1 mM PMSF and 2 µl RNase inhibitor followed by sonication for cell lysis and DNA fragmentation. The cell lysates (supernatants) were collected after centrifugation. Aliquots of 1% lysates were diluted with buffer A and kept at −80 °C to serve as the input control (1% of input RNA). The remaining lysates were incubated with α-FLAG M2 magnetic agarose beads (Sigma, Cat# M8823) on a rotating wheel for 2 h at 4 °C for immunoprecipitation. The bead slurry was recovered using a magnetic stand, washed twice with 1 ml of high-salt wash buffer (50 mM Tris pH 7.4, 1 M NaCl, and 1 mM EDTA), then washed once with 1 ml of IP wash buffer (50 mM Tris pH 7.4, 10 mM MgCl₂, and 0.2% Tween-20). Aliquots of 5% beads were saved for immunoblotting to examine the immunoprecipitation efficiency. The remaining beads were subjected to DNase treatment and the protein–RNA complexes were eluted with 3× FLAG peptide according to the manufacturer's protocol. All samples including 1% input RNA samples were treated at 70 °C for 45 min to reverse cross-linking in buffer B (50 mM Tris pH 7.4, 10 mM DTT, 1% SDS, 1 mg ml⁻¹ protease K, and 5 mM EDTA). Enriched RNAs were extracted using Trizol, precipitated using isopropyl alcohol with linear acrylamide and washed twice with 75% ethanol. Enrichment of the target transcripts was quantified by qPCR using specific primer sets listed in Supplementary Table 2. Transcript enrichment was calculated based on the input RNA (1% of total RNA used for each RIP experiment). The housekeeping gene gyrA was used as a nonspecific negative control. This experiment was performed twice with three biological replicates each time. Statistical analysis was carried out by the paired Student's *t* test using three biological replicates.

**Electrophoretic mobility shift assay (EMSA).** Part of *hitR* coding sequence (237 bp from the start codon) was amplified by PCR using a specific primer set carrying a T7 promoter (Supplementary Table 2) and in vitro RNA transcription was carried out with 1 µg DNA template using a HiScribe™ T7 Quick High Yield RNA Synthesis Kit (NEB) according to the manufacturer's instructions. The in vitro-transcribed RNA was then subjected to dephosphorylation and 5′-end-labeling with (γ-³²P) ATP using KinaseMax 5′ End-Labeling Kit (ThermoFisher). After labeling, G10 columns (NucAway spin columns, Invitrogen) were used to remove the unincorporated ATP. Labeled RNA was denatured for 2 min at 95 °C, chilled on ice (5 min) and renatured by slowly cooling down to room temperature. The EMSA binding assay was set up as follows: 200 ng of labeled RNA probe, varied concentration of KrrA protein, 50 µg ml⁻¹ yeast tRNA, and 1X binding buffer (50 mM Tris-HCl, pH 8.0, 10% glycerol, 300 mM NaCl, 50 µg ml⁻¹ BSA). The reactions were incubated at room temperature for 20 min and subjected to electrophoresis in a 5% Mini-PROTEAN TBE Gel. After electrophoresis, the gels were dried using a gel dryer, exposed to a phosphorimager screen overnight, and scanned by a phosphor image analyzer (Typhoon FLA 7000).

**RNA sequencing and data analysis.** Cells were grown at 30 °C in LB medium overnight and subcultured at a 1:100 ratio into fresh LB medium. After 6 h of growth at 37 °C in the presence or absence of 50 µM '205, aliquots of 1 ml of cell culture were harvested by centrifugation. Total RNA was extracted using an RNeasy Mini Kit following the manufacturer's instructions (Qiagen Sciences, Germantown, MD) and treated with Turbo-DNA free DNase (Ambion). The DNase was removed using DNase removal reagents (Ambion). RNA sequencing was performed with three biological replicates per condition at VANTAGE. The RNA quality was assessed using the 2100 Bioanalyzer (Agilent). At least 200 ng of DNase-treated total RNA with an RNA integrity number greater than 8 was used for rRNA depletion and cDNA library preparation using the ScriptSeq complete (bacterial) kit (Illumina/Epicentre) following the manufacturer's protocol. The cDNA Libraries were assessed using the 2100 Bioanalyzer (Agilent), and quantitated using KAPA library quantification kits (KAPA Biosystems), then subjected to paired-end sequencing on a HiSeq3000 with 150 cycles at VANTAGE.

Bioinformatic analysis was performed using CLC genomic workbench software (version 20.0.1) with default settings. Paired-end sequencing datasets were imported into CLC Genomics Workbench and trimmed to remove barcodes and adapter sequences. To computationally deplete remaining rRNA reads, the entirety of generated reads was aligned to a sequence list containing all rRNA sequences for the genome of *B. anthracis* Sterne (NC_000964). Unmapped (non-rRNA) reads were collected and utilized for RNA-Seq analysis with default settings. A per-sample library size normalization was performed using a TMM method (trimmed mean of M values)[54]. Comparisons of gene expression patterns under different conditions were carried out using the "Differential Expression in Two Groups" function.

**Sporulation efficiency quantification.** *B. anthracis* WT and ΔkrrA were streaked onto LB agar plate and single colonies were used to inoculate LB and grown for 4 h at 37 °C with shaking. Cell cultures were back-diluted with 1:20 ratio into a flask containing modified G medium[55], grown for 24 h at 37 °C with vigorous shaking, and harvested by centrifugation. Aliquots of cell suspensions were serially diluted and enumerated by CFU for total bacterial counts. Meanwhile, the same cell suspensions were heated at 65 °C for 30 min to eliminate all the vegetative cells,

washed with sterile water, and enumerated for surviving spore counts. The sporulation efficiency was calculated by dividing the number of spores by the total number of bacterial cells. Experiments were repeated three times with three biological replicates each time.

**Quantification of spore germination**. *B. anthracis* WT and Δ*krrA* spores were prepared and enumerated as described above. A spore suspension containing $1 \times 10^8$ spores was inoculated into 1 ml of LB medium and incubated at room temperature for 5 min. Aliquots of cell culture were serially diluted and enumerated for total bacterial counts. Meanwhile, the same cell cultures were heated at 65 °C for 30 min to eliminate all the vegetative cells, washed with sterile water, and aliquots of cell suspension were serially diluted and enumerated for viable spore counts. The germination efficiency was calculated by dividing the number of vegetative cells by the total number of cells. The germination efficiency of WT was set as 100%. Three batches of spores prepared independently were used and experiments were carried out three times.

**Quantification of ethanol content**. *B. anthracis* WT cells were grown at 30 °C in LB medium overnight and subcultured at a 1:100 ratio into fresh LB medium amended with 0 or 50 μM '205. After 6 h of growth at 37 °C, cells were harvested by centrifugation and the spent media were filtered through a 0.22-μm membrane filter. The ethanol content was quantified using an ethanol assay kit (cat# MAK076, Sigma-Aldrich, MO) according to the manufacturer's instructions. All samples and standards were run in duplicate and fluorescence intensity ($\lambda_{ex} = 535$ nm and $\lambda_{em} = 587$ nm) was monitored for 30 min. Experiments were carried out three times with three biological replicates each time.

**Reporting summary**. Further information on research design is available in the Nature Research Reporting Summary linked to this article.

## Data availability
The RNA-seq and RIP-seq data that support the findings of this study have been deposited in GEO repository with the accession number GSE193212. The RNA-seq dataset can be found in this link https://www.ncbi.nlm.nih.gov/geo/query/acc.cgi?acc=GSE193210 and the RIP-seq dataset can be found in this link https://www.ncbi.nlm.nih.gov/geo/query/acc.cgi?acc=GSE193211. Source data are provided with this paper.

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

## Acknowledgements

We thank members of the Skaar Laboratory for critical comments on the manuscript. This work was supported by the following grants: National Institutes of Health grants R01 AI73843 (E.P.S.) and R01 AI145992 (E.P.S.), T32 HL094296 (H.P.), F32 AI161860 (H.P.), F32 (AI157215) (A.W.), and T32 ES007028 (C.L.L). H.K.L., X.I.Y., and D.L.S. were supported by the Grove City College Swezey Fund and the Jewell, Moore, and MacKenzie Fund.

## Author contributions

H.P. and E.P.S. conceived and designed the experiments. A.W. processed the fRIP-seq data and created the Circus plot, C.L.L performed the XylE assay, C.M.G. performed the whole-genome sequencing analysis, H.K.L., X.I.Y., and D.L.S. constructed *B. anthracis* strains, and H.P. performed all other experiments. H.P. drafted the paper. H.P. and E.P.S. edited the paper. All authors reviewed the paper.

## Competing interests

The authors declare no competing interests.
