## [Peer Review File · Nature Communications]

Reviewers' Comments:

Reviewer #1:

Remarks to the Author:

Pi et al. present a manuscript entitled "A *Bacillus anthracis* RNA binding protein post-transcriptionally regulates two-component signaling through RNA turnover".

In this study, the authors identified KrrA as a modulator of the activity the HitRS two-component that acts by modulating mRNA stability.

This manuscript is very well written and contains a huge amount of well performed and solid experiments. However, I have major problems with some of the conclusions that are not fully supported by the data presented and validation experiments that are missing.

Major comments

1) the screen to identify regulators of the HitRS system is nice. However, it would be important to explain (if it is the case) that the basal expression level of the hitPRS operon is very low. Was a particular genetic context used for the screen?

2) It is difficult to understand why the authors identified the KrrA suppressor since the regulation was only two-fold when the Hit promoter was tested with a fusion to *xylE*. Where there additional mutations identified by sequencing in the suppressor strain ?

3) On line 132, the authors mention the HitRS activator '205 (already published) molecule that is used during the whole study. They should explain what this activator does and justify its use in this study.

4) The whole study is missing a $\Delta krrA$ mutant that is complemented by a wild type *krrA* copy.

5) I find no data that convincingly support the model of KrrA influencing mRNA stability.

- the differences in half-lives presented in figure 2 are very small (probably not significant) and the corresponding northern blots are not shown. In addition, on line 146, the authors use a "representative suppressor B26" to measure mRNA stability. This measurement should be done with the clean deletion mutant as well as with the complemented mutant.

- if mRNA stability would be modified, it should be detected by RT Q-PCR and it is not. Why?

- there are no in vitro data showing that KrrA has strong affinity to RNA (or DNA).

6) There is no validation of the KrrA -mediated regulation by western blot showing that the expression of at least one of its targets is really changed (this has been provided in the PlosGenet paper with ComK in figure 3C).

7) Several important controls are missing in the CLIP assay experiments intended to detect the KrrA-interaction RNA complexes. These are (i) the same experiment with cross linking on a strain only expressing the Flag-tag and (ii) more importantly, immunoprecipitation under conditions without cross-linking. Why was the CLIP method used in the paragraph starting at line 168 and the formaldehyde cross-linking for the following paragraph, again immuno-precipitated RNA without cross-linking should be analyzed as a control.

8) In many examples, the conclusions are extrapolated and not fully supported by the data, f.i, on line 216 "the findings demonstrate that KrrA acts as a pleiotropic post-transcriptional RNA regulator..", same problem on line 230-231, on line 254: "Notably, KrrA directly binds to eight transcripts" the direct binding is not demonstrated, on line 300 "KrrA regulation altered the RNA levels of 73 direct targets.."

9) Figure 4, given the high number of upregulated genes in the KrrA mutant, one can wonder whether there could be a bias due to normalization. Was it normalized to the whole RNA seq data?

10) in fact, the KrrA is a very close homolog of the Kre protein (a ComK repressor) of the closely related organism *Bacillus subtilis* (60% identity with the *B. anthracis* protein) that is described in details in Gamba et al. Plos Genet 2015. In that publication, figure 7 already identified the gene homologous protein of *B. anthracis*. So, the finding of KrrA is not really novel.

11) coIP of KrrA-Flag, is the list presented in Table 1 complete? Controls should also be performed in the presence of RNase and of DNase, to see whether the purified partners are not just associated through an RNA or DNA molecule. Validation by two hybrid could reinforce the data.

Minor comments

- in Table S3, what are the other mutated alleles obtained during the screen?

- on line 158, I guess it is predicted rather than predicated?

- line 353-354, what do you mean by "in times of cell envelope disruption"?

Reviewer #2:

Remarks to the Author:

Pi et al. report about an analysis of the expression of the HitRS two-component system in *B. anthracis*. The authors demonstrate that HitR-dependent expression of the hit operon is increased upon inactivation of KrrA. Based on the lack of an effect on transcription initiation and on the assignment of the similar *B. subtilis* Kre protein as a mediator of RNA degradation, the authors show that KrrA binds to many RNA molecules including the hit mRNA and suggest that this triggers degradation of the target RNA by recruitment of the RNA degradosome.

Major comments

- #1 It would be highly desirable to see data on the mechanism of RNA degradation: is it degradation or processing (as the part on differential expression in the Discussion suggests). Which RNase is responsible? Does KrrA interact with this RNase? It would really be essential to go deeper into the mechanism by which KrrA controls hit mRNA stability.
- #2 From the ms, it is not clear whether KrrA is a regulator (then it should respond to some environmental changes) or whether it is just an effector protein that always helps to set the hitR:hitS mRNA ratio! In the latter case, it should not be termed regulator.
- #3 What is the effect of KrrA inactivation in a strain lacking the RNase responsible for hit mRNA degradation/ processing?
- #4 The ms is often difficult to read as it lacks information or the presentation is extremely confusing. This may be eligible for lab members, but not for the average interested reader. Some examples are given below.
- #5 The interactions between KrrA and other proteins probably involved in RNA processing should be validated by complementary experimental approaches.

Specific comments:

- #1 l. 34: is there any evidence for a RNA degradosome in *B. anthracis*?
- #2 l. 35: in response to what kind of changes, please specify!
- #3 l. 63: you may add that TCS can be controlled by binding of second messengers as well (e.g. KdpD in Gram-positive bacteria by c-di-AMP)!
- #4 l. 86: What is Kre? Please explain briefly for the non-specialist.
- #5 l. 132: What is '205? This needs an introduction, it is completely unclear to the broad audience of the journal.
- #6 l. 134: either "vehicle or '205 treated conditions": What is vector? This is only eligible to extreme insiders but not to the broad readership of Nat. Comms!
- #7 l. 128 ... 144: shorten to one or two sentences and move the text to the supplement.
- #8 l. 142: Again, the rationale behind the '205 treatment is not clear.
- #9 l. 153: better "suggest". Better present Northern blots as this would allow to distinguish between degradation and processing!
- #10 l. 158: better "predicted"
- #11 l. 165: "These results indicate .. " It cannot be excluded that KrrA requires some cofactor for nuclease activity. Therefore, as generally with negative results, one should be more cautious with conclusions. Better "These results suggest"
- #12 l. 170: better "no previously characterized RNA-binding domain"
- #13 l. 193: As a control, better use a tagged protein that does not bind RNA.
- #14 l. 214: "implicating KrrA plays", better "indicating that KrrA plays"
- #15 l. 230: Sometimes the authors speak of a hitRS operon, and use its promoter (directly upstream of hitR, right?), sometimes it is hitPRS. This is really confusing! Moreover, to which portion of the mRNA does KrrA bind, and which portion is affected by the loss of KrrA. This is all unclear and makes the paper really difficult to follow. What is the function of the hitP gene product?
Please be consistent with hitPRS vs hitRS vs hitR!
- #16 l. 230: "establishing", better "suggesting" or "indicating". There is no evidence presented that KrrA indeed facilitates target degradation!
- #17 l. 254/255: KrrA binds to 10 sporulation/germination mRNAs. Is the expression of these genes affected by the loss of KrrA? Please make a clear statement!

- #18 l. 276/277: Are there three distinct mRNAs for SR1P in B. anthracis?
- #19 l. 277: not clear how Fig. 5C helps here!
- #20 l. 264 ... 318: This part is somewhat off-topic. Better move to the supplement!
- #21 l. 331: Please make statements on enrichment factors!
- #22 l. 333: it should be mentioned that SbcD is a DNA nuclease.
- #23 l. 353: does KrrA bind to the hit PRS transcript or only to the hitR part of it?
- #24 l. 358: What is meant by "fundamentally uncoupled"
- #25 l. 368: differential expression: this could easily be addressed by Northern blots.
- #26 l. 403: Why does the sentence start with "While"? What does it refer to?
- #27 l. 412: check spelling of "anthracis"

RESPONSE TO REVIEWER COMMENTS

REVIEWER 1

Reviewer one stated that “*This manuscript is very well written and contains a huge amount of well performed and solid experiments.*” However, this Reviewer had concerns that some of the conclusions were not fully supported by the presented data and additional data in support of this model were requested. We have adding significant new experimentation and text edits to address these points.

Major comments

1) the screen to identify regulators of the HitRS system is nice. However, it would be important to explain (if it is the case) that the basal expression level of the *hitPRS* operon is very low. Was a particular genetic context used for the screen?

The strain we used for the genetic selection is a *B. anthracis* WT strain harboring the erythromycin resistance gene *ermC* driven by the *hitPRS* promoter (WT $P_{hitermC}$), which is integrated into a pseudogene locus *bas3009*. We added text in line 99-101 describing the specific genetic context of this strain. The basal expression of *hitPRS* is very low as shown in our prior study (PMID: 33362282) as well as in this study (Figure 2A). The power of the assay is its sensitivity that allows the discovery of regulatory factors that would be difficult to detect using conventional selections/screens. This is a key strength of the manuscript.

2) It is difficult to understand why the authors identified the KrrA suppressor since the regulation was only two-fold when the Hit promoter was tested with a fusion to *xyIE*. Where there additional mutations identified by sequencing in the suppressor strain?

There were no mutations in the genome of the suppressor strains other than those found in *krrA*. This highlights the sensitivity of this genetic selection and underscores the power of the assay since it allowed us to identify KrrA as a new factor involved in gene regulation.

3) On line 132, the authors mention the HitRS activator '205 (already published) molecule that is used during the whole study. They should explain what this activator does and justify its use in this study.

We added text in line 138-140 to better describe '205 and its utility for the experiments described in this study.

4) The whole study is missing a $\Delta krrA$ mutant that is complemented by a wild type *krrA* copy.

The original submission included both a $\Delta krrA$ and a complemented strain $\Delta krrA$ pOS1 $P_{igt.krrA}$ -FLAG, which was used for CLIP and fRIP-seq experiments. To test the functionality of the FLAG-tagged KrrA, we introduced this construct into $\Delta krrA$ $P_{hitermC}$ and checked the erythromycin resistance phenotype. This complemented strain substantially reverted the erythromycin resistance phenotype as seen in $\Delta krrA$ $P_{hitermC}$ (Figure S3), indicative of successful genetic complementation. In our original

experiments we did not observe full complementation, therefore we performed whole genome sequencing and confirmed that there is no secondary mutation in this complemented strain. We edited the text in line 127-128 and moved the data to Figure 1C-D to highlight this complementation experiment.

5) I find no data that convincingly support the model of KrrA influencing mRNA stability. - the differences in half-lives presented in figure 2 are very small (probably not significant) and the corresponding northern blots are not shown. In addition, on line 146, the authors use a "representative suppressor B26" to measure mRNA stability. This measurement should be done with the clean deletion mutant as well as with the complemented mutant.

We appreciate the Reviewer's concern regarding the magnitude of the impact of KrrA on mRNA stability and we have addressed this question by performing the suggested Northern blot. These data are included in panel A of a new version of Figure 2. We found that the *hitPRS* transcript level is extremely low and only detectable in $\Delta krrA$ treated with the inducer '205. Due to the low abundance of *hitPRS* transcript, a more sensitive method of qPCR was employed to determine the effects of KrrA on *hitPRS* mRNA stability. This was done in two sets of strains: (i) a representative suppressor B26 vs its parental WT strain (Figure 2B-C) and (ii) the clean deletion mutant $\Delta KrrA$ vs WT (Figure 2D-E). All these data are presented in Figure 2 and we added text in line 154-159 describing these results. Combined, these data show that KrrA plays a critical role in modulating *hitPRS* transcript stability.

- if mRNA stability would be modified, it should be detected by RT Q-PCR and it is not. Why?

We observed a 2-fold increase of *hitPRS* mRNA levels in $\Delta krrA$ compared to WT as quantified by qPCR (Figure S3D). This was done with three independent experiments using three biological replicates each time. We also observed a 3-fold significant increase of *hitR* in RNAseq (Data Table S6). We highlighted this point in the revised manuscript (line 142-143).

- there are no in vitro data showing that KrrA has strong affinity to RNA (or DNA).

The Reviewer makes an excellent point. To address this concern, we performed EMSAs to evaluate KrrA binding to its target *hitR* *in vitro*. These new data are included in a revised version of Figure S7C-D, and clearly demonstrate that KrrA has a strong affinity to *hitR* transcript *in vitro*. We added text in line 247-252 describing these results.

6) There is no validation of the KrrA -mediated regulation by western blot showing that the expression of at least one of its targets is really changed (this has been provided in the PlosGenet paper with ComK in figure 3C).

Thank you for this suggestion. To address this concern, we evaluated the impact of KrrA on HitR and HitS. We constructed Myc tagged constructs for HitR and HitS that are driven by the native *hit* promoter: pOS1. P_{hit} *hitR*-myc and pOS1. P_{hit} *hitS*-myc. The pOS1. P_{hit} *hitR*-myc construct did not result in detectable protein so was not pursued further. However, pOS1. P_{hit} *hitS*-myc worked well. We introduced the pOS1. P_{hit} *hitS*-myc construct into both WT and $\Delta krrA$ and performed Western blotting

to examine the impact of KrrA on HitS protein abundance using anti-Myc antibody. We found that HitS expression is significantly elevated in $\Delta krrA$ compared to WT, particularly upon '205 treatment, indicating that KrrA-mediated RNA regulation impacts protein expression of its targets. The new data are presented as Figure S5 and text describing these results is in line 167-169.

7) Several important controls are missing in the CLIP assay experiments intended to detect the KrrA-interaction RNA complexes. These are (i) the same experiment with cross linking on a strain only expressing the Flag-tag and (ii) more importantly, immunoprecipitation under conditions without cross-linking. Why was the CLIP method used in the paragraph starting at line 168 and the formaldehyde cross-linking for the following paragraph, again immuno-precipitated RNA without cross-linking should be analyzed as a control.

We performed significant new experimentation to better control the experiment, as suggested by the Reviewer. We repeated the CLIP experiment under two conditions (in the presence or absence of '205 treatment) with or without cross-linking. We also included an untagged WT strain with or without cross-linking as negative controls to evaluate nonspecific background and ensure specificity. These new data are included in a revised version of Figure 3A. The data clearly demonstrate that KrrA directly binds to some RNAs even without cross-linking regardless of '205 treatment. As anticipated, cross-linking drastically enhanced the recovery of co-immunoprecipitated RNAs. We added text in line 195-198 describing these results. We hope the Reviewers agree that the CLIP experiments are now well-controlled.

To clarify why the CLIP method was used in the paragraph starting at line 168 and the formaldehyde cross-linking for the following paragraph, we used UV-crosslinking for CLIP experiments but formaldehyde crosslinking for RIP-seq in the next paragraph. One of the advantages of formaldehyde as a crosslinking reagent is that the formaldehyde cross-links are reversible, allowing us to reverse the cross-links and harvest RNAs for high throughput sequencing (fRIP-seq).

8) In many examples, the conclusions are extrapolated and not fully supported by the data, f.i, on line 216 "the findings demonstrate that KrrA acts as a pleiotropic post-transcriptional RNA regulator..", same problem on line 230-231, on line 254: "Notably, KrrA directly binds to eight transcripts" the direct binding is not demonstrated, on line 300 "KrrA regulation altered the RNA levels of 73 direct targets.."

We altered the text to address these points and temper our conclusions.

9) Figure 4, given the high number of upregulated genes in the KrrA mutant, one can wonder whether there could be a bias due to normalization. Was it normalized to the whole RNA seq data?

Yes, we understand the sequencing depth may differ between samples, a per-sample library size normalization was performed using a TMM method (trimmed mean of M values) (Robinson, M. D. and A. Oshlack (2010). "A scaling normalization method for differential expression analysis of RNA-seq data." *Genome Biology* 11(3): R25). Libraries sizes are then used as part of the per-sample normalization. We have now included text changes in line 642-644 and line 713-714 to describe how

we performed normalization in the Methods section. We also like to point out that the transcript of *krrA* is 2489-fold lower in $\Delta krrA$ compared to WT under vehicle-treated conditions while 3433-fold lower in $\Delta krrA$ compared to WT under '205-treated conditions, the magnitude of this change skews the dataset towards a high number of upregulated genes in $\Delta krrA$ compared to WT. However as shown in Figure 4A-B, the majority of the upregulated genes are only altered 3- to 20-fold.

10) in fact, the KrrA is a very close homolog of the Kre protein (a ComK repressor) of the closely related organism *Bacillus subtilis* (60% identity with the *B. anthracis* protein) that is described in details in Gamba et al. Plos Genet 2015. In that publication, figure 7 already identified the gene homologous protein of *B. anthracis*. So, the finding of KrrA is not really novel.

We contend that a protein that is only 60% identical across two very similar organisms suggests evolutionary pressures have driven distinct functions between these two proteins. In addition, *B. anthracis* is an understudied yet important organism, so the identification of a protein that has a global impact on gene expression is a significant finding. Finally, our manuscript describes a large and thorough characterization of the RNA targets of KrrA, far beyond what was done in the single paper describing Kre in *B. subtilis* which we predict will lead to considerable new discoveries and be of broad interest. We altered the text in line 108-112 to address these points.

11) coIP of KrrA-Flag, is the list presented in Table 1 complete? Controls should also be performed in the presence of RNase and of DNase, to see whether the purified partners are not just associated through an RNA or DNA molecule. Validation by two hybrid could reinforce the data.

We appreciate the reviewer's comment and understand it is of critical importance to determine how KrrA affects RNA stability mechanistically. However, we think this is beyond the scope of this study. We plan to follow up on this and further validate the putative KrrA-interacting protein partners using genetic and/or biochemistry approaches: either inactivate those genes to test their role in KrrA-dependent protein production or test their interaction between KrrA and RNases *in vitro* using purified recombinant proteins.

Minor comments

- in Table S3, what are the other mutated alleles obtained during the screen?

We have included all the mutants isolated from genetic selection in Table S3.

- on line 158, I guess it is predicted rather than predicated?

We altered the text.

- line 353-354, what do you mean by "in times of cell envelope disruption"?

We altered the text to address this.

Reviewer #2 (Remarks to the Author):

Pi et al. report about an analysis of the expression of the HitRS two-component system in B. anthracis. The authors demonstrate that HitR-dependent expression of the hit operon is increased upon inactivation of KrrA. Based on the lack of an effect on transcription initiation and on the assignment of the similar B. subtilis Kre protein as a mediator of RNA degradation, the authors show that KrrA binds to many RNA molecules including the hit mRNA and suggest that this triggers degradation of the target RNA by recruitment of the RNA degradosome.

Major comments

#1 It would be highly desirable to see data on the mechanism of RNA degradation: is it degradation or processing (as the part on differential expression in the Discussion suggests). Which RNase is responsible? Does KrrA interact with this RNase? It would really be essential to go deeper into the mechanism by which KrrA controls hit mRNA stability.

As stated in response to Reviewer 1, we performed Northern blot as requested and the data are included in panel A of Figure 2. We found that the *hitPRS* transcript level is extremely low and only detectable in $\Delta krrA$ treated with the inducer '205. However, due to the low abundance of *hitPRS* transcript, we were unable to distinguish the role of KrrA between RNA degradation and processing. However, it clear that KrrA plays a critical role in modulating *hitPRS* transcript stability which is a significant finding.

We appreciate the Reviewer's comment and understand it is of critical importance to determine how KrrA affects RNA stability mechanistically. However, we think that identifying additional proteins involved in this process is beyond the scope of this study.

#2 From the ms, it is not clear whether KrrA is a regulator (then it should respond to some environmental changes) or whether it is just an effector protein that always helps to set the hitR:hitS mRNA ratio! In the latter case, it should not be termed regulator.

We have edited the text so that KrrA is not termed a regulator.

#3 What is the effect of KrrA inactivation in a strain lacking the RNase responsible for hit mRNA degradation/ processing?

We appreciate the Reviewer's comment and understand it is of critical importance to determine how KrrA affects RNA stability mechanistically. However, we do not know the identity of the RNase responsible for degradation/processing of these mRNAs in *B. anthracis*.

#4 The ms is often difficult to read as it lacks information or the presentation is extremely confusing. This may be eligible for lab members, but not for the average interested reader. Some examples are given below.

The Reviewers were conflicted on whether the manuscript was well “*very well written*”, or “*extremely confusing*”. Therefore, we were judicious regarding the changes we made to the text. We tried very hard to ensure clarity throughout the document.

#5 The interactions between KrrA and other proteins probably involved in RNA processing should be validated by complementary experimental approaches.

We appreciate the Reviewer’s comment and understand it is of critical importance to understand how KrrA affects RNA stability mechanistically. However, we think it is beyond the scope of this study.

Specific comments:

#1 l. 34: is there any evidence for a RNA degradosome in *B. anthracis*?

An RNA degradosome has not been discovered in *B. anthracis*, however homologues to known systems are present in the genome.

#2 l. 35: in response to what kind of changes, please specify!

We have edited the text.

#3 l. 63: you may add that TCS can be controlled by binding of second messengers as well (e.g. KdpD in Gram-positive bacteria by c-di-AMP)!

We edited the text and added additional reference in line 61 describing c-di-AMP regulating transcriptional output of *kdpD*.

#4 l. 86: What is Kre? Please explain briefly for the non-specialist.

We have edited the text to better describe Kre.

#5 l. 132: What is ‘205? This needs an introduction, it is completely unclear to the broad audience of the journal.

We have edited the text in line 138-140 to include new information that hopefully improves clarity.

#6 l. 134: either “vehicle or ‘205 treated conditions”: What is vector? This is only eligible to extreme insiders but not to the broad readership of Nat. Comms!

We have edited the text in 140-141 to clarify this.

#7 l. 128 ... 144: shorten to one or two sentences and move the text to the supplement.

We have made the recommended changes.

#8 l. 142: Again, the rationale behind the '205 treatment is not clear.

We have edited the text in line 138-140 to better explain the rationale for using '205.

#9 l. 153: better "suggest". Better present Northern blots as this would allow to distinguish between degradation and processing!

We have made the suggested text change and performed Northern blots as requested and the data are included in a new panel A of Figure 2. These experiments show that the *hitPRS* transcript level is extremely low and only detectable in $\Delta krrA$ treated with the inducer '205. Due to the low abundance of *hitPRS* transcript, we were unable to distinguish the role of KrrA between RNA degradation and processing. However, it is clear that KrrA plays a critical role in modulating *hitPRS* transcript stability which is the major conclusion of this manuscript.

#10 l. 158: better "predicted"

We edited the text.

#11 l. 165: "These results indicate .. " It cannot be excluded that KrrA requires some cofactor for nuclease activity. Therefore, as generally with negative results, one should be more cautious with conclusions. Better "These results suggest"

We modified the text in line 182-183 to highlight this point. We agree with the Reviewer on this point, and the requirement for an as-yet-unidentified cofactor may be exactly the reason why some of their other requested experiments are challenging to complete.

#12 l. 170: better "no previously characterized RNA-binding domain"

We edited the text.

#13 l. 193: As a control, better use a tagged protein that does not bind RNA.

As stated in response to Reviewer 1, we have now included new controls and repeated the CLIP experiment under two conditions (in the presence or absence of '205 treatment) with or without cross-linking. We also included an untagged WT strain with or without cross-linking as negative controls to demonstrate nonspecific background and ensure specificity. The updated data are included in a new version of Figure 3A. These data clearly demonstrate that KrrA directly binds to some RNAs even without cross-linking regardless of '205 treatment. And as anticipated, cross-linking drastically enhanced the recovery of co-immunoprecipitated RNAs. We hope the reviewers agree that the CLIP experiments are now well-controlled.

#14 l. 214: “implicating KrrA plays ...”, better “indicating that KrrA plays”

We edited the text.

#15 l. 230: Sometimes the authors speak of a hitRS operon, and use its promoter (directly upstream of hitR, right?), sometimes it is hitPRS. This is really confusing! Moreover, to which portion of the mRNA does KrrA bind, and which portion is affected by the loss of KrrA. This is all unclear and makes the paper really difficult to follow. What is the function of the hitP gene product? Please be consistent with hitPRS vs hitRS vs hitR!

We have edited the text to address this. The function of HitP is unknown.

#16 l. 230: “establishing”, better “suggesting” or “indicating”. There is no evidence presented that KrrA indeed facilitates target degradation!

We agree with the Reviewer and have edited the text to make this correction.

#17 l. 254/255: KrrA binds to 10 sporulation/germination mRNAs. Is the expression of these genes affected by the loss of KrrA? Please make a clear statement!

We have edited the text in line 278-283 to discuss the RNAseq and RIPseq data more clearly in that section. These results revealed that eight transcripts coding for sporulation proteins (*spoIID*, *spoIIE*, *spoVR*, *spoVM*, *spoVID*, *BAS3655*, *yqfD*, and *yabG*) and two transcripts coding for germination proteins (*gerPA* and *gerPF*) were significantly enriched following KrrA immunoprecipitation. Among these ten transcripts, three of them (*spoVR*, *spoVID*, and *BAS3655*) were significantly elevated in $\Delta krrA$ compared to WT.

#18 l. 276/277: Are there three distinct mRNAs for SR1P in *B. anthracis*?

Yes, there are three distinct mRNAs for SR1P in *B. anthracis*: BAS_RS19665, BAS_RS19665, and BAS_RS19675. The peptides encoded by these three mRNAs share high sequence identity as shown below.

```
SR1Pc    MGTIVCQDCEGTIAHFEDEKVTVLYGKCGS-CGCDHTEHTKAQ    42
SR1Pa    MGTIVCQVCEGTIGHFEDEKSTVLYGKCGSHCDDHKEHKA-      42
SR1Pb    MGTIVCQVCEGTIGHFEDEKTTVLYGKCGTNCDCASKDNAKA-    42
          *****  *****  *****  *****: *.*  .:::**
```

#19 l. 277: not clear how Fig. 5C helps here!

We have modified text in line 291-294 and line 304 to 307 to explain how Figure 5C puts the data into context. Since we used the HitRS activator '205 in many of our experiments, we think it is important to determine the impact of this compound on bacterial physiology and KrrA function. As we illustrated in Figure 5C, 24 genes involved in gluconeogenesis, TCA cycle, and butanoate fermentation were significantly downregulated while *gapA*, encoding the glycolytic enzyme glyceraldehyde 3-

phosphate dehydrogenase, *ldh*, responsible for fermentation of pyruvate to lactate, and *adhE*, involved in ethanol fermentation, were notably upregulated following '205 treatment. And we further examined the effects of '205 on ethanol fermentation, and found that following treatment of '205, the ethanol content in the spent media of *B. anthracis* WT culture was ~4 times higher relative to vehicle. These data indicate that besides *HitRS* activation, '205 extensively impacts bacterial physiology and diverts cellular metabolism from aerobic respiration to fermentation.

#20 l. 264 ... 318: This part is somewhat off-topic. Better move to the supplement!

We edited the text line 291-294 and line 304 to 307 to better integrate this point. It is important to note that *KrrA* only binds to *hitR* transcript under vehicle-treated conditions while *KrrA* binds to the *hitPRS* operon under '205-treated conditions. Therefore, it is critical to define the transcriptomic changes in response to '205 treatment as well as the coregulatory gene networks related to *KrrA*-mediated RNA regulation.

#21 l. 331: Please make statements on enrichment factors!

We agree that the discussion of the potential interaction partners is premature and confusing. We have removed the CoIP results from the manuscript since they require further biochemical and genetic verification in order for us to draw firm conclusions.

#22 l. 333: it should be mentioned that *SbcD* is a DNA nuclease.

As stated above, we have removed the CoIP results from the manuscript since they require further biochemical and genetic verification in order for us to draw firm conclusions.

#23 l. 353: does *KrrA* bind to the *hitPRS* transcript or only to the *hitR* part of it?

Under vehicle-treated conditions, *KrrA* only binds to *hitR* transcript, while under '205-treated conditions, *KrrA* binds to *hitPRS* transcript.

#24 l. 358: What is meant by "fundamentally uncoupled"?

We have edited the text to clarify this.

#25 l. 368: differential expression: this could easily be addressed by Northern blots.

As stated above, we performed Northern blot as requested and the data are included in panel A of Figure 2. We found that the *hitPRS* transcript level is extremely low and only detectable in $\Delta krrA$ treated with the inducer '205. These data indicate that *KrrA* plays a critical role in modulating *hitPRS* transcript stability. Thank you for this suggestion.

#26 l. 403: Why does the sentence start with "While"? What does it refer to?

We have edited the text.

#27 l. 412: check spelling of “anthracis”

We checked the spelling and edited the text.

Reviewers' Comments:

Reviewer #1:

Remarks to the Author:

The authors have convincingly answered to my comments, they made additional experiments and corrected some parts of the manuscripts. The revised version is much improved and deserves publication.

Reviewer #2:

Remarks to the Author:

The authors have added new information, and the presentation is now improved.

The key message of the paper is the involvement of Krr in the control of hitRS mRNA stability. Unfortunately, the data presented in Fig. 2, are not convincing as the effect is very minor, and I have serious doubts concerning the significance.

Fig. S6 should be moved to the main document.

RESPONSE TO REVIEWER COMMENTS

Reviewer #1 (Remarks to the Author):

The authors have convincingly answered to my comments, they made additional experiments and corrected some parts of the manuscripts. The revised version is much improved and deserves publication.

We thank the reviewer for the positive comments.

Reviewer #2 (Remarks to the Author):

The authors have added new information, and the presentation is now improved.

We thank the reviewer for the positive comments.

The key message of the paper is the involvement of Krr in the control of *hitRS* mRNA stability. Unfortunately, the data presented in Fig. 2, are not convincing as the effect is very minor, and I have serious doubts concerning the significance.

We appreciate the Reviewer's concern regarding the magnitude of the impact of KrrA on *hitRS* mRNA stability and we have addressed this question by performing Northern blot as suggested for the first revision. In addition, *B. anthracis* is an understudied yet important pathogen, so the identification of a protein that has a global impact on gene expression is a significant finding. Finally, our manuscript describes a large and thorough characterization of the RNA targets of KrrA, which we predict will lead to considerable new discoveries and be of broad interest.

Fig. S6 should be moved to the main document.

We have moved S6 to the main document as the Reviewer suggested. Figure S6A-B are in a revised version of Figure 3F-G and Figure S6C-D are in a revised version of Figure 4.